# A parabrachial-hypothalamic parallel circuit governs cold defense in mice

Wen Z. Yang[1,13], Hengchang Xie [1,2,3,13], Xiaosa Du[1,13], Qian Zhou[1,2,3,13], Yan Xiao[1], Zhengdong Zhao[1], Xiaoning Jia[1], Jianhui Xu[4], Wen Zhang[2], Shuang Cai[5], Zhangjie Li[1], Xin Fu[1], Rong Hua[6], Junhao Cai[1], Shuang Chang[1], Jing Sun[1], Hongbin Sun[1], Qingqing Xu[7], Xinyan Ni[1], Hongqing Tu[1], Ruimao Zheng [8,9], Xiaohong Xu [2], Hong Wang[10], Yu Fu [11], Liming Wang[12], Xi Li[7], Haitao Yang [1], Qiyuan Yao[6], Tian Yu[5]✉, Qiwei Shen [6]✉ & Wei L. Shen [1]✉

Thermal homeostasis is vital for mammals and is controlled by brain neurocircuits. Yet, the neural pathways responsible for cold defense regulation are still unclear. Here, we found that a pathway from the lateral parabrachial nucleus (LPB) to the dorsomedial hypothalamus (DMH), which runs parallel to the canonical LPB to preoptic area (POA) pathway, is also crucial for cold defense. Together, these pathways make an equivalent and cumulative contribution, forming a parallel circuit. Specifically, activation of the LPB→DMH pathway induced strong cold-defense responses, including increases in thermogenesis of brown adipose tissue (BAT), muscle shivering, heart rate, and locomotion. Further, we identified somatostatin neurons in the LPB that target DMH to promote BAT thermogenesis. Therefore, we reveal a parallel circuit governing cold defense in mice, which enables resilience to hypothermia and provides a scalable and robust network in heat production, reshaping our understanding of neural circuit regulation of homeostatic behaviors.

Hypothermia caused by cold or famine is one of the major death causes in wild animals and has been a threat to human life throughout history[1–3]. Hypothermia reduces motor function and suppresses immune responses, thereby impairing behavioral reactions and increasing infection risks, respectively[4,5]. Thus, it is vital to maintain a stable core body temperature ($T_{core}$) or to rapidly recover from a hypothermic state. In response to cold exposure, the brain initiates a series of countermeasures to defend $T_{core}$ from hypothermia, including increases in the thermogenesis of brown adipose tissue (BAT), skeletal muscle shivering, heart rate (HR), and skin vasoconstriction. Several behavioral adaptations are also induced[6]. However, the neural circuitry underlying these cold-defense activities is not well understood.

[1]Shanghai Institute for Advanced Immunochemical Studies & School of Life Science and Technology, Shanghaitech University, Shanghai 201210, China. [2]Institute of Neuroscience, State Key Laboratory of Neuroscience, CAS Center for Excellence in Brain Science and Intelligence Technology, Shanghai Institutes for Biological Sciences, Chinese Academy of Sciences, Shanghai 200031, China. [3]University of Chinese Academy of Sciences, Beijing 100049, China. [4]Thermoregulation and Inflammation Laboratory, Chengdu Medical College, Chengdu, Sichuan 610500, China. [5]Guizhou Key Laboratory of Anesthesia and Organ Protection, Zunyi Medical University, Zunyi 563006, China. [6]Department of General Surgery, Huashan Hospital, Fudan University, Shanghai 200433, China. [7]Institute of life sciences, Chongqing Medical University, Chongqing 400044, China. [8]Department of Anatomy, Histology and Embryology, Health Science Center, Peking University, Beijing 100871, China. [9]Neuroscience Research Institute, Peking University, Beijing 100871, China. [10]Brain Cognition and Brain Disease Institute (BCBDI), Shenzhen Institutes of Advanced Technology, Chinese Academy of Sciences; Shenzhen-Hong Kong Institute of Brain Science-Shenzhen Fundamental Research Institutions, Shenzhen 518055, China. [11]Institute of Molecular and Cell Biology, Agency for Science, Technology and Research (A*STAR), Singapore 138667, Singapore. [12]Institute of Molecular Physiology, Shenzhen Bay Laboratory, Shenzhen 518055, China. [13]These authors contributed equally: Wen Z. Yang, Hengchang Xie, Xiaosa Du, Qian Zhou. ✉e-mail: zzyutian@126.com; shenqiwei@huashan.org.cn; shenwei@shanghaitech.edu.cn

The hypothalamic preoptic area (POA) is the known thermo-regulation center in mammals[6–10], and remarkable advances have been made in dissecting POA circuits that mediate warm defense[11–22]. On contrary, the circuitry related to the POA in cold defense remains elusive. Previous studies[23] in rats have suggested that GABAergic neurons in the median preoptic nucleus (MnPO) are critical for cold defense, which receive cold sensory input from the external lateral LPB (LPBel)[24,25] and suppress GABAergic neurons in the medial preoptic area (MPA) to disinhibit cold-defense responses. However, in contrast to rat studies, optogenetic activation of the LPB→POA pathway in mice results in hypothermia alone[25,26], and activation of GABAergic neurons in the MnPO or MPA in mice could not increase $T_{core}$ (hyperthermic)[13,14,19,27]. Therefore, whether the LPB→POA pathway could function in cold defense in mice is still not clear. Interestingly, recent studies in mice suggested that selective activation of GABAergic arginine vasopressin (AVP) neurons in the MPA or Bombesin-like receptor 3 (Brs3) neurons (mixed glutamatergic and GABAergic) in the MnPO and ventromedial preoptic nucleus (VMPO) could induce hyperthermia[28,29], although this induced hyperthermia is rather mild (~1°C)[30]. Noticeably, it is still debatable the existence of AVP neurons in the POA[20,31]. Together, the unresolved cold-defense function of the LPB→POA pathway and the mild hyperthermic responses induced by POA neurons in mice raise a concern that the POA may contribute less to cold defense than previously thought. POA-independent mechanisms may also exist to boost cold defense (at least) in mice.

The dorsomedial hypothalamus (DMH) is considered as a thermogenesis center that functions downstream of the POA[10]. DMH lesions selectively impair thermoregulation under cold but not warm challenges[32]. Studies in rats have suggested that DMH neurons are excited or disinhibited by POA cold-defense neurons to increase thermogenesis[23,33]. Opposite to what is seen in the POA, activation of most thermoregulatory neural types in the DMH triggers hyperthermia (by up to 1.8°C)[14,34–37] in mice. To date, only neurons expressing choline acetyltransferase (ChAT) in the DMH, have been shown to induce mild hypothermia (−1°C)[38]. Since diverse types of DMH neurons could induce stronger hyperthermia than POA neurons, it raises an interesting possibility that the DMH may function in parallel to the POA – rather than solely downstream of the POA – to boost cold defense[28,33,39]. In retrospect, tracing studies have shown that the LPB sends neural projections directly to the DMH in mice[35,40]. Therefore, we hypothesized that the LPB directly transmits cold signals to the DMH to boost cold defense cooperatively the LPB→POA pathway.

To test our hypothesis, we first determined the connectivity, neural activity, and function of the LPB→DMH pathway in cold defense. We found that activating this pathway increased $T_{core}$ and promoted a wide spectrum of cold defense activities while blocking this pathway led to profound cold intolerance. This pathway functioned in parallel with the LPB→POA pathway, in that the two pathways provide an equivalent and cumulative contribution to cold defense. Using projection-specific transcriptomic analysis, we identified a somatostatin[+] (SST[+]) neural subpopulation in the LPB that targets the DMH to promote BAT thermogenesis. Thus, our findings reveal that parabrachial-hypothalamic parallel pathways function cooperatively to promote cold defense in mice, thereby providing a parallel circuit model for understanding thermoregulation.

## Results

### The LPB→POA/DMH pathways are sensitive to cold temperature

To test our hypothesis that the LPB→DMH pathway functions independently of the POA in cold defense, we mapped axonal projection patterns of LPB glutamatergic neurons, the predominant neuronal subtype in the LPB. This was accomplished by injecting AAVs carrying Cre-dependent ChR2-eYFP (AAV9-DIO-ChR2-eYFP) into the LPB of Vglut2-IRES-Cre mice (LPB[Vglut2] neurons) (Fig. 1a, b and Supplementary

Fig. 1). As expected, LPB[Vglut2] neurons projected to many regions, including the POA (mostly the MnPO and VMPO) and the DMH. To assess potential collateral projections between axons within the POA and the DMH, we simultaneously injected retrograde AAVs carrying Cre-dependent GFP (Retro-DIO-GFPL10) and Cre-dependent mCherry (Retro-DIO-mCherry) into the POA and the DMH of Vglut2-IRES-Cre mice, respectively (Fig. 1c). Three weeks post-injection, these mice were exposed to cold (10°C, 2 h) or warm (38°C, 2 h) stimuli and then sacrificed for cFos immunostaining to evaluate the thermal responsiveness of neurons. We verified that GFPL10 expression was limited to the POA (mostly MnPO and VMPO), and that mCherry expression was limited to the DMH (Supplementary Fig. 2a, b). POA-projecting (green) and DMH-projecting (red) LPB[Vglut2] neurons were more concentrated in the external lateral LPB (LPBel) and dorsal LPB (LPBd) where temperature-responsive neurons reside[24,25] (Fig. 1d). Among all warm-activated LPB neurons (cFos[+]; 100%), 40% projected to the POA, 13% projected to the DMH, and 7% projected to both regions (Fig. 1e, f and Supplementary Fig. 2c). These results suggest that warm-activated LPB neurons mainly innervate the POA compared to the DMH.

Among cold-activated LPB neurons (cFos[+]; 100%), 45% projected to the POA, 38% projected to the DMH, and 20% projected to both regions (Fig. 1g, h and Supplementary Fig. 2d). These results suggest that a similar ratio of cold-activated LPB neurons innervates the POA or DMH, and that there are substantial collateral projections between these projection axons. We further analyzed the temperature responsiveness of DMH-projecting LPB neurons. About 30% responded to cold, whereas only 10% responded to warmth (Fig. 1i), suggesting a biased response to cold temperature.

To verify whether LPB[Vglut2] neurons directly innervate DMH neurons, we injected AAVs carrying Cre-dependent ChR2 into the LPB of Vglut2-Cre mice and recorded light-induced excitatory postsynaptic currents (EPSCs) from DMH neurons while photostimulating ChR2-expressing neural terminals in the DMH projected from LPB[Vglut2] neurons (Fig. 1j). Light stimulations faithfully induced EPSCs and could be blocked by the glutamate receptor antagonists CNQX and AP5 (Fig. 1j, middle), indicating a glutamatergic transmission. Latency after photoactivation was within the range of a monosynaptic connection (Fig. 1j, right)[41]. Notably, nearly 33% of recorded DMH neurons (17/51 neurons from three mice; randomly selected) displayed EPSCs to light stimulations (Fig. 1k), suggesting that the DMH receives dense inputs from the LPB.

### Neural activity dynamics of the LPB→DMH pathway in response to temperature

To visualize neural activity dynamics of the LPB[Vglut2]→DMH pathway in response to temperature stimuli, we used the calcium reporter GCaMP6s and fiber photometry to record calcium dynamics[42]. We first recorded calcium dynamics at LPB[Vglut2 & GCaMP6s] terminals in the DMH, where these terminals were in close proximity to cold-induced cFos (Fig. 2a, b). Floor temperature ($T_{floor}$) was controlled by a Peltier device (Fig. 2a). As expected, these terminals displayed larger responses to floor cooling (25→10°C) compared to floor warming (25→38°C) (Fig. 2c–e).

To decode the responses to cold temperatures, we measured calcium dynamics during a series of temperature steps with the same starting $T_{floor}$ (25°C) and cooling rate, but ending at different $T_{floor}$, namely 4, 10, 16, or 21°C (Fig. 2f). Interestingly, we observed robust calcium responses during the cooling phases (referred as cooling response) for all four temperature steps (Fig. 2g). Yet, peak values of these cooling responses are similar (Fig. 2h), suggesting that calcium responses do not encode cooling amplitudes. These results are consistent with the responses seen in cold-responsive spinal cord relay neurons[43]. As the $T_{floor}$ stabilized, neuronal responses gradually reduced (or adapted to the steady temperature). This adaptive feature is also consistent with patterns seen in the majority (~70%) of

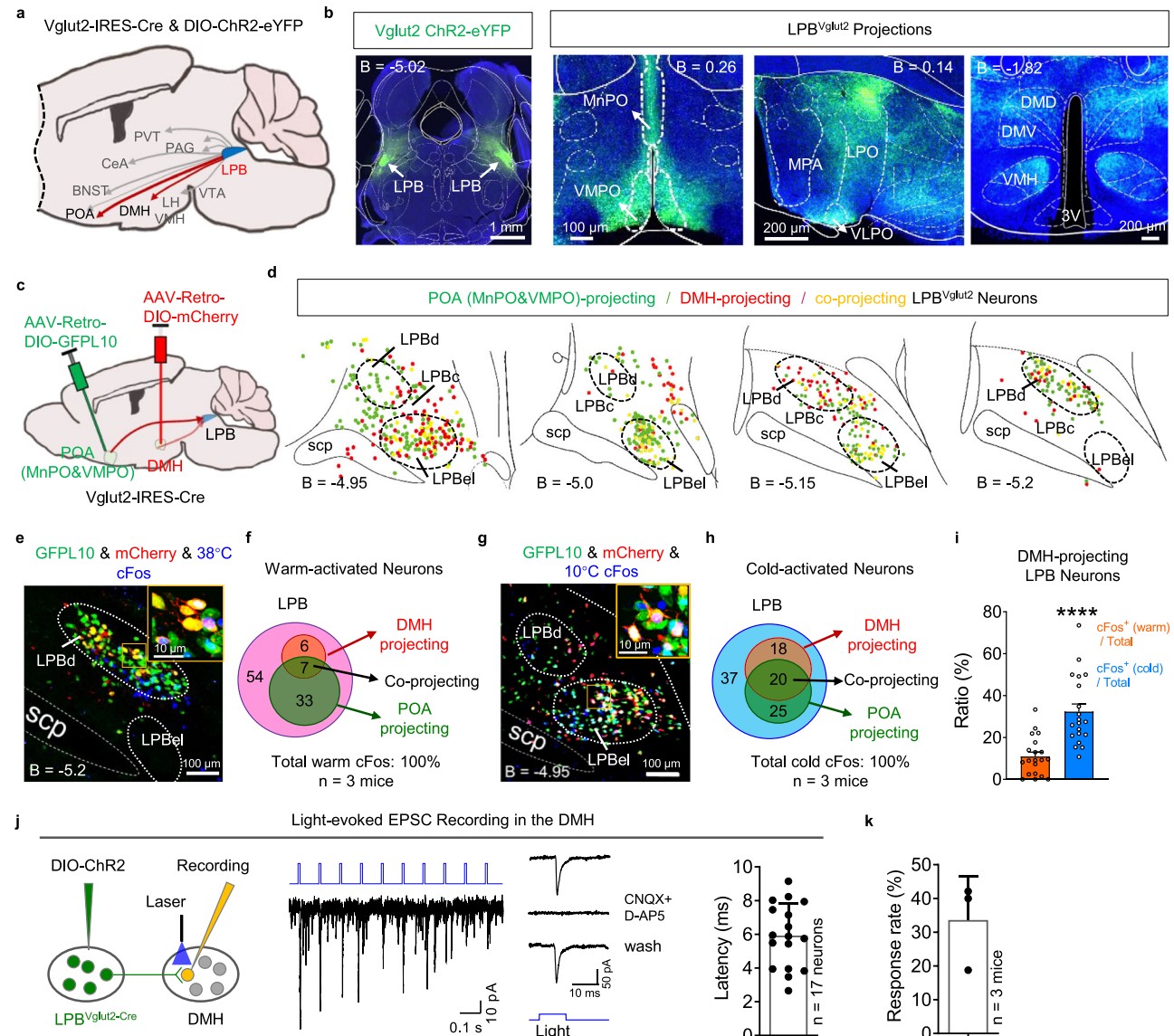

**Fig. 1 | Properties of the LPB → POA/DMH pathways and their sensitivity to temperature. a, b** Whole-brain projection pattern of LPB$^{Vglut2}$ neurons viewed using AAV-mediated axonal tracing. The injection site and representative axonal images in the POA and the DMH are shown in (**b**). The scale bar is with the image (as with all brain images). B, bregma; BNST, bed nucleus of the stria terminalis; CeA, central nucleus of the amygdala; 3 V, third ventricle; LPB, lateral parabrachial nucleus; PAG, periaqueductal gray matter; VTA, ventral tegmental area; LPO, lateral preoptic area; LH, lateral hypothalamus; PVT, paraventricular thalamic nucleus; MPA, medial preoptic area; MnPO, median preoptic area; VLPO, ventrolateral preoptic area; VMPO, ventromedial preoptic nucleus; VMH, ventromedial hypothalamic nucleus; POA, preoptic area; DMH, dorsomedial hypothalamic nucleus; DMV, dorsomedial hypothalamic nucleus, ventral part; DMD, dorsomedial hypothalamic nucleus, dorsal part. This experiment was repeated at least 4 times independently with similar results. **c, d** Mapping POA-projecting and DMH-projecting LPB$^{Vglut2}$ neurons. LPBc, lateral parabrachial nucleus, central part; LPBd, lateral parabrachial nucleus,

dorsal part; LPBel, lateral parabrachial nucleus, external and internal part; scp, superior cerebellar peduncle. **e–h** Overlap between DMH/POA-projecting LPB$^{Vglut2}$ neurons and warmth (**e, f**) or cold (**g, h**) induced cFos. The experiments in (**e**) and (**g**) were repeated 3 times independently with similar results, respectively. **i** The ratio of warm- and cold-activated DMH-projecting LPB$^{Vglut2}$ neurons (20 slices each from 3 mice). **j** Excitatory postsynaptic currents (EPSCs) were recorded randomly from DMH neurons after photoactivation of LPB$^{Vglut2 \& ChR2}$ terminals in the DMH (blue, 7 mW, 10 ms). EPSCs were blocked by GluR antagonists D-AP5 and CNQX. The mean latency of responsive neurons is shown on the right ($n = 17$ neurons). **k** The EPSC response rate of DMH neurons to light stimulation ($n = 3$ mice). A total of 17 responsive neurons out of 51 randomly recorded DMH neurons were from three mice. All data are the mean ± sem, and were analyzed using the unpaired t-test in (**i**). $P$-values were calculated based on statistical tests in Supplementary Table 2. ****$p \leq 0.0001$. Source data are provided as a Source data file.

peripheral thermosensory neurons[44]. Next, we quantified mean responses during the steady phase (referred as cold response) and found there were no differences between the responses at 16 and 21°C (Fig. 2i). Yet, as T$_{floor}$ became less than 10°C, cold responses increased significantly with colder T$_{floor}$ (10 or 4°C) (Fig. 2i). Thus, the LPB → DMH projection is sensitive to different cool/cold temperatures, where 10°C appears to be the turning point for faster increases in cold responses.

This projection is also responsive to cooling but not sensitive to different cooling amplitudes.

We further measured calcium dynamics in response to four T$_{floor}$ cooling steps, with the same cooling rate and drop in temperature ($\Delta T = -15$°C) but different starting temperatures (from 25 to 35°C) (Fig. 2j). These cooling steps induced similar peak cooling responses (Fig. 2j, k), suggesting an insensitivity to starting/ending

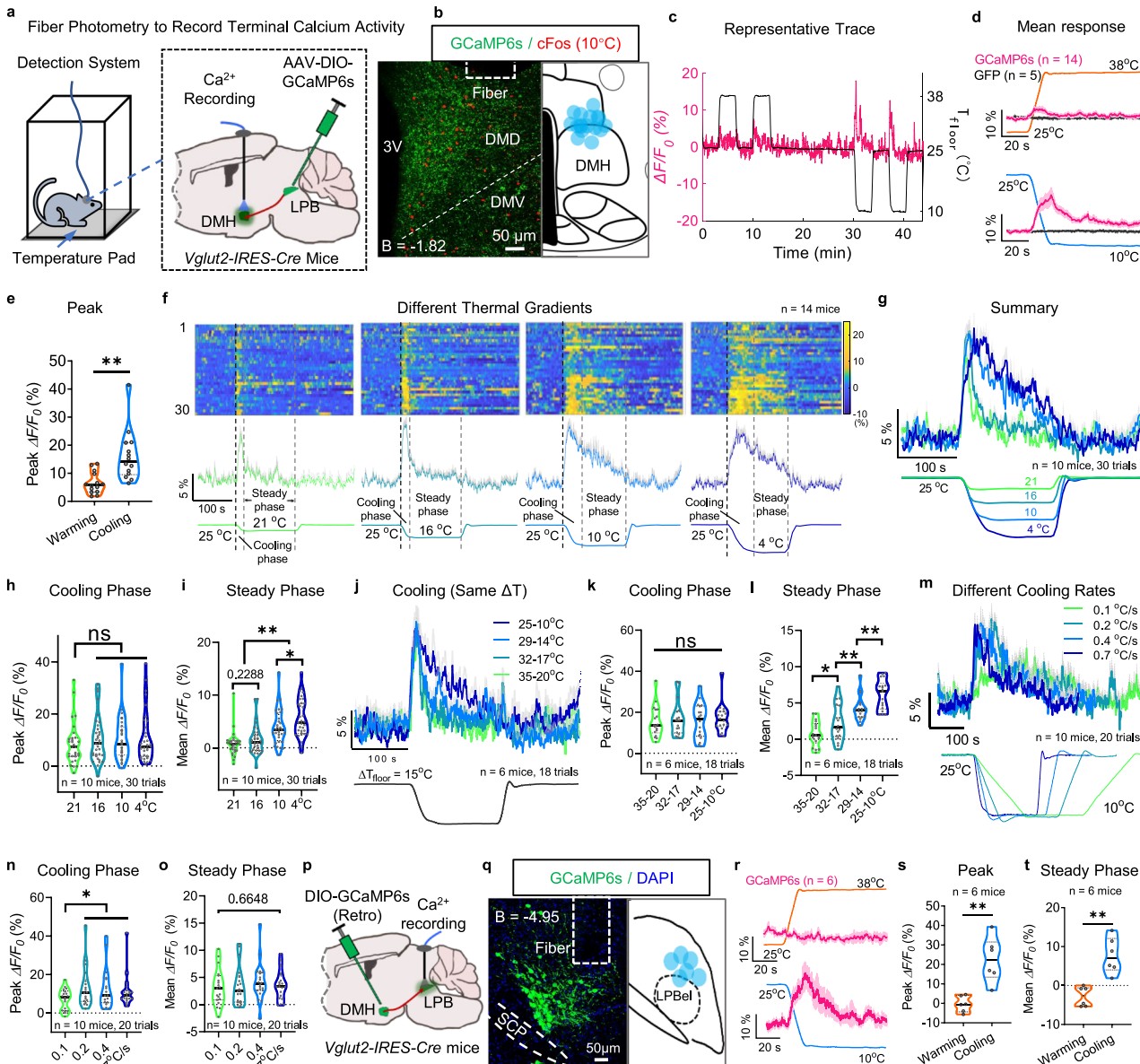

**Fig. 2 | Neural dynamics of LPB → DMH projection in response to thermal stimuli. a** Fiber photometry to record the calcium dynamics of LPB$^{Vglut2}$ neural terminals in the DMH with GCaMP6s. **b** Representative expression of GCaMP6s from LPB$^{Vglut2}$ neural terminals (left) and summary of fiber tracts (shown as blue dots, right) in the DMH with cold-induced cFos immunoactivity. This experiment was repeated at least 14 times independently with similar results. **c** A representative trace showing calcium dynamics of LPB$^{Vglut2}$ neural terminals in the DMH in response to warm and cold stimuli. **d, e** Calcium dynamics after warming or cooling the floor. (GCaMP6s, $n = 14$ mice; GFP, $n = 5$ mice). **f–i** Calcium dynamics (**f, g**) and quantification (**h, i**) of LPB$^{Vglut2}$ neural terminals in the DMH in response to different cool/cold floor temperatures. **j–l** Calcium dynamics (**j**) and quantification (**k, l**) of LPB$^{Vglut2}$ neural terminals in the DMH in response to cooling from different starting temperatures but the same temperature reduction. ($n = 6$ mice, 3–5 trials per

mouse). **m–o** Calcium dynamics (**m**) and quantification (**n, o**) of LPB$^{Vglut2}$ neural terminals in the DMH in response to different cooling rates. ($n = 10$ mice, 2 trials per mouse). **p, q** Recording from DMH-projecting LPB$^{Vglut2}$ neurons (**p**) and representative expression of GCaMP6s (left) and summary of fiber tracts (shown as blue dots, right) (**q**). This experiment was repeated at least 6 times independently with similar results. **r** Calcium dynamics of DMH-projecting LPB$^{Vglut2}$ neurons in response to floor warming or cooling ($n = 6$ mice). **s, t** Peak ΔF/F$_0$ values (**s**) during floor warming or cooling and Mean ΔF/F$_0$ values (**t**) during the steady phase after floor warming or cooling ($n = 6$ mice). All data are the mean ± sem (except **c**), and (**h, i, k, l, n, o**) were analyzed by one-way RM ANOVA followed by Bonferroni's multiple comparisons test, (**e, s, t**) were analyzed by unpaired t-test. P-values are calculated based on statistical tests in Supplementary Table 2. *$p \leq 0.05$; **$p \leq 0.01$; ns, not significant. Source data are provided as a Source data file.

temperatures. However, the cold responses increased with colder ending temperatures (Fig. 2l). These data suggest that cold responses, rather than cooling responses, are sensitive to cold temperature values.

To characterize responses to different cooling rates, we measured calcium dynamics in response to a series of cooling rates when cooling from 25 to 10°C. Calcium signals increased in similar dynamics as the temperature changes (Fig. 2m), but no peak differences were detected

between the three higher cooling rates (0.2, 0.4, or 0.7°C/s) (Fig. 2n), which is consistent with the responses seen in spinal cord relay neurons[43]. However, peak responses were smaller at a much slower cooling rate (0.1°C/s) (Fig. 2n), suggesting a very weak sensitivity to cooling rates. As the ending temperatures were the same, the cold responses were not different (Fig. 2o). Hence, these results suggest that cooling responses of LPB$^{Vglut2}$ terminals in the DMH are weakly sensitive to cooling rates.

Finally, we sought to verify the sensitivity of the LPB$^{Vglut2}$ → DMH pathway to cold temperatures by recording calcium levels within the soma of DMH-projecting LPB$^{Vglut2}$ neurons (Fig. 2p). We injected retrograde traveling AAVs carrying Cre-dependent GCaMP6s into the DMH to drive the expression of GCaMP6s in the LPB of Vglut2-Cre mice (Fig. 2q). As expected, these DMH-projecting LPB$^{Vglut2}$ neurons were selectively activated by cooling (25 → 10°C) and cold temperatures (10°C) compared to warming (25 → 38°C) or warm temperatures (38°C) (Fig. 2r–t), suggesting a biased sensitivity to cooling/cold temperatures. Together, we show that the LPB → DMH pathway is responsive to both cooling and cold temperatures. The cold responses are sensitive to different cold temperatures. Yet, akin to spinal cord relay neurons, the peak cooling responses are not sensitive to temperature values and cooling amplitudes, and are only weakly sensitive to cooling rates.

### DMH-projecting LPB neurons that do not collaterally project to the POA (DMH$^{only}$-projecting) are sufficient to mediate cold defense

After confirming the biased response of the LPB → DMH pathway to cool/cold temperatures, we sought to evaluate the function of the LPB → DMH/POA pathways in cold defense. As half of the DMH-projecting cold-responsive LPB neurons also sent collateral projections to the POA (Fig. 1h and Supplementary Fig. 3), we, therefore, tested the cold-defense function of LPB neurons that projected only to the DMH (DMH$^{only}$-projecting) and compared their function to LPB neurons that project only to the POA (POA$^{only}$-projecting). To do so, we used the Cre_off & FlpO_on strategy[45] to express neurotoxin TeNT (Fig. 3a, b and Supplementary Fig. 4a, b). On day 1, we injected retrograde AAVs carrying Cre (AAV-Retro-hSyn-Cre) into the POA (MnPO and VMPO) and AAVs carrying Cre_off & Flp_on TeNT into the LPB, where Cre would retrogradely travel to the LPB to turn off TeNT expression (Cre_off). Turning off TeNT expression unblocked POA-projecting LPB neurons. On day 8, we injected retrograde AAVs carrying FlpO (AAV-Retro-hSyn-FlpO) into the DMH to turn on the expression of TeNT in the LPB (FlpO_on), thus blocking DMH$^{only}$-projecting LPB neurons (Fig. 3a and Supplementary Fig. 4a). The GFP carried in AAV9-hSyn-Cre_off-FlpO_on-eGFP was used as the control for TeNT (Fig. 3a). Using the same strategy, we also blocked POA$^{only}$-projecting LPB neurons (Fig. 3b and Supplementary Fig. 4b). To validate the specificity of this strategy, we co-injected AAV9-hSyn-DIO-mCherry with AAV9-Cre_off & FlpO_on-TeNT-eGFP into the LPB, where mCherry labels POA-projecting LPB neurons after recombination by Cre traveling from the POA and TeNT-GFP labels DMH$^{only}$-projecting LPB neurons. The low amount of overlap (<2%) between mCherry (POA-projecting) and TeNT-eGFP (DMH$^{only}$-projecting) indicated that this strategy was specific (Supplementary Fig. 4c). The overlap between DMH-projecting and POA$^{only}$-projecting neurons was also very low (<2%; Supplementary Fig. 4d).

$T_{core}$ and physical activity were measured by telemetry probes implanted intraperitoneally. Under a thermoneutral condition (29°C), blocking POA$^{only}$-projecting or DMH$^{only}$-projecting LPB$^{Vglut2}$ neurons did not affect body weight, basal $T_{core}$, energy expenditure (EE), or physical activity compared with controls (Supplementary Fig. 4e–h). To evaluate cold defense phenotypes, we switched ambient temperature ($T_a$) from 29 to 22, 16, 10, or 4°C. When the $T_a$ was dropped to 22, 16, or 10°C, blocking these neurons caused significant but similar degrees of hypothermia (Fig. 3c–e). When $T_a$ was dropped to 4°C, however, blocking POA$^{only}$-projecting neurons resulted in slightly more severe hypothermia than when DMH$^{only}$-projecting neurons were blocked (Fig. 3f). This suggests that DMH$^{only}$-projecting and POA$^{only}$-projecting LPB neurons function independently and contribute equally to defense responses to cool/cold temperatures (>10°C), while POA$^{only}$-projecting LPB neurons contribute more to defense responses in cold/nociceptive temperatures (4°C).

### The LPB → POA/DMH pathways form a parallel circuit in cold defense

After showing that the LPB → DMH pathway could function independently of the POA, we tested whether the LPB → POA/DMH pathways function as parallel circuits where these two pathways function independently and cumulatively in cold defense. Thus, we used TeNT to block neurons downstream of the LPB, including LPB-innervating POA neurons, LPB-innervating DMH neurons, or both (Fig. 3g and Supplementary Fig. 5a). We injected anterograde transsynaptic AAV1-hSyn-Cre[46] into the LPB and Cre-dependent TeNT into the POA (MnPO, VMPO, MPA, and lateral preoptic area (LPO)), the DMH, or both. These blocked LPB-innervating POA neurons (POA$^{LPB}$ blocking), LPB-innervating DMH neurons (DMH$^{LPB}$ blocking), or both (co-blocking) (Fig. 3g and Supplementary Fig. 5a). Injecting Cre-dependent GFP into both the POA and the DMH served as the control (GFP control). TeNT expression heatmaps are shown in Supplementary Fig. 5b. Under thermoneutral conditions (29°C), no changes in basal $T_{core}$ were seen for any of the blocking groups (Supplementary Fig. 5c, d, g). Physical activity did not change in the DMH$^{LPB}$ blocking and co-blocking groups (Supplementary Fig. 5f, h), but POA$^{LPB}$ blocking mice slightly increased physical activity in the light phase and basal EE during both the light and dark phases (Supplementary Fig. 5e, f).

Under cool/cold conditions (29 → 22/16/10/4°C switches), all blocking groups showed substantial hypothermia compared to the control (Fig. 3h–k). Notably, there were no differences in the hypothermia between DMH$^{LPB}$ blocking and POA$^{LPB}$ blocking groups, which is reminiscent of phenotypes observed after blocking LPB input neurons (Fig. 3c–f). However, the co-blocking group always exhibited more substantial hypothermia, suggesting a cumulative or additive effect (Fig. 3h–k). Hence, these results demonstrate that the LPB → POA/DMH pathways provide an equivalent and cumulative contribution to cold defense, forming a parallel circuit to maintain $T_{core}$.

We further performed the cold plate test and found no differences in withdrawal latency or the number of lifts between DMH$^{LPB}$ blocking, POA$^{LPB}$ blocking, and controls (Supplementary Fig. 5i), suggesting these pathways do not affect cold nociception. In the temperature preference tests, where mice could choose between 30°C and 6, 10, 16, or 35°C for 5 minutes, no preference differences were detected between these groups (Supplementary Fig. 5j). Together, these data suggest that the LPB → POA/DMH pathways form parallel circuits to coordinate cold defense. They play a negligible role in cold nociception or thermotactic behaviors.

### The LPB$^{Vglut2}$ → DMH pathway induces strong hyperthermia

Having shown that the LPB → DMH pathway is necessary for cold defense, we sought to test whether this pathway is sufficient to elicit cold defense responses. We injected AAVs carrying Cre-dependent ChR2 into the LPB of Vglut2-Cre mice and photostimulated ChR2-expressing LPB$^{Vglut2}$ neural terminals in the DMH (Fig. 4a, b and Supplementary Fig. 6). A 10-Hz photoactivation of LPB$^{Vglut2}$ neural terminals in the DMH elevated $T_{core}$ by 1.81°C, which was accompanied by an abrupt increase in physical activity (Fig. 4c, d). Noticeably, 5 or 20-Hz photoactivation did not show a significant difference in $T_{core}$ increases (Supplementary Fig. 7a, b). To estimate the amplitudes of antidromic propagation of action potentials, we recorded the soma calcium activity and did not find any calcium signal changes in LPB somata due to DMH terminal photoactivation (Supplementary Fig. 7c), suggesting a weak effect in the antidromic propagation of action potentials. Nevertheless, to rule out antidromic propagation of action potentials during terminal stimulation, we chemogenetically inhibited LPB$^{Vglut2}$ somas via hM4D$_i$ while photostimulating their terminals in the DMH (Fig. 4e, f). We confirmed the efficacy of soma inhibition via calcium fiber photometry (Supplementary Fig. 7d). This inhibition did not affect the induced hyperthermia, suggesting the hyperthermic effect was specific to terminal activation (Fig. 4g).

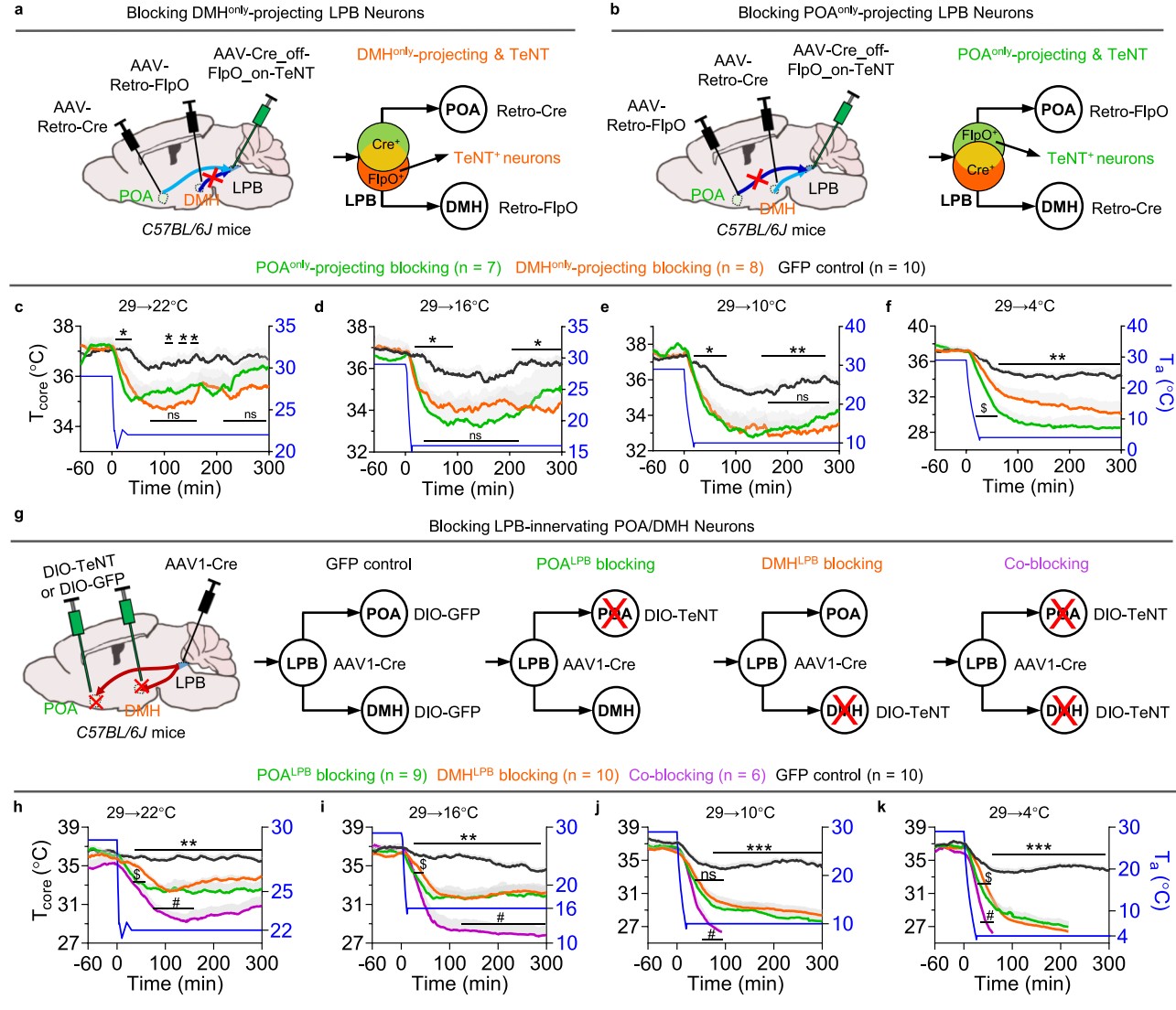

**Fig. 3 | The LPB → POA and LPB → DMH pathways provide an equivalent and cumulative contribution to cold defense. a** Blocking LPB neurons that project to the DMH but do not send collateral projections to the POA (DMH^only-projecting LPB neurons). See also Supplementary Fig. 4a for a detailed description. **b** Blocking LPB neurons that project to the POA but do not send collateral projections to the DMH (POA^only-projecting LPB neurons). See also Supplementary Fig. 4b for a detailed description. **c–f** $T_{core}$ changes in response to a series of cold exposures, namely 29 → 22°C (**c**), 29 → 16°C (**d**), 29 → 10°C (**e**), and 29 → 4°C (**f**), after blocking POA^only-projecting LPB neurons or DMH^only-projecting LPB neurons. DMH^only-projecting TeNT, $n = 8$ mice; POA^only-projecting TeNT, $n = 7$ mice; GFP controls, $n = 10$ mice. *, GFP vs. DMH^only-projecting blocking group. $, DMH^only projecting vs. POA^only

projecting blocking. **g** Blocking LPB-innervating POA/DMH neurons or both using TeNT. The detailed scheme is shown in Supplementary Fig. 5a. **h–k** $T_{core}$ changes in response to a series of cold exposures, namely 29 → 22°C (**h**), 29 → 16°C (**i**), 29 → 10°C (**j**), and 29 → 4°C (**k**), after blocking POA^LPB, DMH^LPB, or both types of neurons. $n = 10$ mice for GFP and DMH^LPB blocking group; $n = 9$ mice for POA^LPB blocking group; $n = 6$ mice for the co-blocking group. *, DMH^LPB blocking vs. GFP; $, DMH^LPB vs. POA^LPB blocking; #, POA^LPB vs. co-blocking. All data are the mean ± SEM, and (**c–f**, **h–k**) were analyzed by two-way RM ANOVA followed by uncorrected Fisher's LSD test. The *p*-values are calculated based on statistical tests in Supplementary Table 2. *$p ≤ 0.05$; **$p ≤ 0.01$; ***$p ≤ 0.001$; $$p ≤ 0.05$; #$p ≤ 0.05$; ns, not significant. Source data are provided as a Source data file.

## Cold-defense responses induced by activating the LPB^Vglut2 → DMH pathway depend on DMH^Vglut2/Vgat neurons but not POA neurons

To identify DMH neuronal cell types targeted by the LPB, we used the anterograde transsynaptic AAV1 tracer to label LPB-innervating DMH neurons and co-stained with GABAergic and glutamatergic markers in the DMH. To do so, we simultaneously injected AAV1-hSyn-FlpO into the LPB, and a mixture of AAVs carrying FlpO-dependent mCherry (AAV8-FDIO-mCherry) and Cre-dependent GFP (AAV9-DIO-GFP) into the DMH, in either Vglut2-IRES-Cre or Vgat-IRES-Cre mice (Fig. 4h). Interestingly, the mCherry (LPB-innervating DMH neurons) overlapped with both Vgat^+ ( ~ 60%) and Vglut2^+ markers ( ~ 20%) (Fig. 4i), suggesting that both GABAergic and glutamatergic DMH neurons are innervated by the LPB.

To determine which DMH neural types are required for the hyperthermic function of the LPB^Vglut2 → DMH pathway, we blocked DMH^Vglut2/Vgat neurons using TeNT while photoactivating LPB^Vglut2 & ChR2 terminals in the DMH (Fig. 4j and Supplementary Fig. 7e–n). The Vgat-T2A-FlpO used for labeling DMH^Vgat neurons was validated by GABA immunostaining (Supplementary Fig. 7i). As a comparison, we also blocked POA neurons using TeNT expressed in the MnPO, VMPO, MPA, and LPO (Fig. 4j and Supplementary Fig. 7k–n). As expected, blocking DMH^Vglut2, DMH^Vgat, or POA neurons rendered mice intolerant to cold exposure (4°C) (Supplementary Fig. 7g,j,m); this is consistent with the known functions of these neurons[14]. Interestingly, blocking either DMH^Vglut2 or DMH^Vgat neurons abolished the hyperthermia induced by photoactivating LPB^Vglut2 terminals in the DMH (Fig. 4k). In contrast, POA blocking did not affect this induced hyperthermia (Fig. 4k).

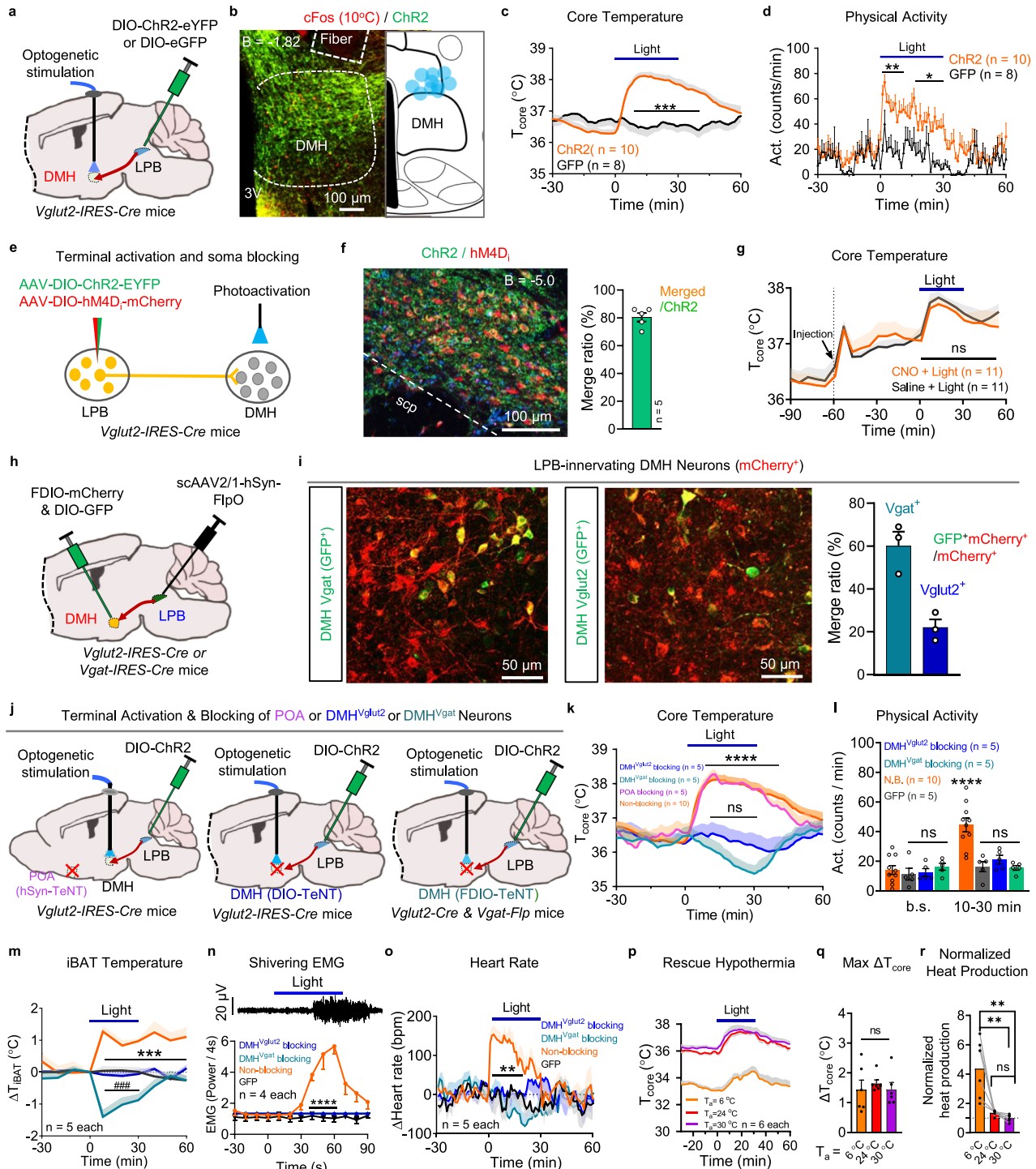

**Fig. 4 | Activation of the LPB$^{Vglut2}$ → DMH projection induces strong cold defense responses independent of the POA. a** Activating the LPB$^{Vglut2}$ → DMH projection via optogenetics. **b** Expression of ChR2 from LPB$^{Vglut2}$ neural terminals (left) and summary of fiber tracts (shown as blue dots, right) in the DMH. This experiment was repeated at least 10 times independently with similar results. **c, d** Changes of T$_{core}$ (**c**) and physical activity (**d**) after photoactivation of LPB$^{Vglut2 \& ChR2}$ neural terminals in the DMH (ChR2, n = 10 mice; GFP, n = 8 mice). **e** Activating the LPB$^{Vglut2}$ → DMH projection while blocking LPB somas with hM4D$_i$. **f** Representative ChR2 and hM4Di co-expression in LPB$^{Vglut2}$ neurons (n = 5 mice). **g** Changes in T$_{core}$ after activating the LPB$^{Vglut2}$ → DMH projection while blocking LPB neurons (*n* = 11 mice). (CNO, i.p., 10 mg/kg). **h** The neurotransmitter properties of LPB-innervating DMH neurons. **i** The overlap between LPB-innervating DMH neurons (mCherry$^+$) and GABAergic or glutamatergic markers (GFP$^+$). (n = 3 mice each). **j, k** Schemes for photoactivation (**j**) and changes in T$_{core}$ (**k**) after photoactivation of the

LPB$^{Vglut2}$ → DMH projection while blocking downstream neurons. (n = 5 mice each). **l–o** Blocking of DMH$^{Vglut2}$ or DMH$^{Vgat}$ neurons abolished photoactivation-induced increases in physical activity (**l**), iBAT thermogenesis (**m**), nuchal muscle shivering EMG (**n**), and heart rate (**o**). Animal numbers were indicated. **p–r** Changes in T$_{core}$ (**p**), maximum ΔT$_{core}$ (**q**) and normalized heat production (**r**) after photoactivation of the LPB$^{Vglut2}$ → DMH projection under different T$_a$ (*n* = 6 mice each). Heat production was normalized by the formula (T$_{core}$ -T$_a$) / ΔT$_{core}$ as reported[47]. All data are the mean ± sem, and were analyzed by two-way RM ANOVA followed by Bonferroni's multiple comparisons tests (**c, g, k–o**), or by uncorrected Fisher's LSD test (**d**), (**q** and **r**) were analyzed by paired t-test. The p-values are calculated based on statistical tests in Supplementary Table 2. *$p \le 0.05$; **$p \le 0.01$; ***$p \le 0.001$; ****$p \le 0.0001$; ###$p \le 0.001$; ns, not significant. Light pattern: 473 nm, 12 mW, 10 Hz, 10 ms, 2-s on 2-s off, duration as indicated. Source data are provided as a Source data file.

Unexpectedly, photoactivation caused mild hypothermia after blocking DMH[Vgat] neurons (Fig. 4k), which may be caused by activating unknown hypothermic neural axons. Therefore, the hyperthermic effect of the LPB[Vglut2]→DMH pathway depends on DMH[Vglut2/Vgat] neurons but not POA neurons.

To further determine which autonomic cold-defense responses are recruited by the LPB[Vglut2]→DMH pathway, we measured physical activity and heart rate by telemetry probes, interscapular BAT temperature ($T_{iBAT}$) by infrared thermography, and muscle shivering by nuchal muscle electromyography (EMG). Interestingly, photoactivation was sufficient to increase physical activity, $T_{iBAT}$, shivering EMG, and heart rate (Fig. 4l–o). Blocking either DMH[Vglut2] or DMH[Vgat] neurons abolished these responses, suggesting that both neural types are essential for these defense responses. Like $T_{core}$ changes, $T_{iBAT}$ decreased in response to photoactivation after blocking DMH[Vgat] neurons (Fig. 4m).

### Activation of the LPB[Vglut2]→DMH pathway rapidly relieves hypothermia

As activation of the LPB[Vglut2]→DMH pathway induced robust hyperthermic responses, we wondered whether it could alleviate cold-induced hypothermia. Notably, photoactivation of LPB[Vglut2 & ChR2] terminals in the DMH substantially relieved the hypothermia observed after placing mice at $T_a$ of 6°C (Fig. 4p). Surprisingly, $T_{core}$ increases were similar for mice placed at $T_a$ of 6, 24, or 30°C (Fig. 4q), even though heat loss was much greater at lower $T_a$[47]. Therefore, we normalized heat production by considering the heat-loss effects and found that heat production increased 4-fold at $T_a$ of 6°C compared to that seen at $T_a$ of 24°C (Fig. 4r). These results collectively suggest that the LPB[Vglut2]→DMH pathway robustly restores $T_{core}$ in mice subjected to cold.

### Activation of the LPB[Vglut2]→DMH pathway increases iBAT thermogenesis and suppresses body weight gain

An increase in $T_{iBAT}$ may indicate BAT thermogenesis or may only mirror the increase in $T_{core}$. To differentiate between these possibilities, we increased the temporal resolution in $T_{core}$ and $T_{iBAT}$ recording using custom-made wired probes[25] (Fig. 5a). After photoactivating LPB[Vglut2]→DMH projections, the rise of $T_{iBAT}$ preceded $T_{core}$ and reached a higher value than $T_{core}$ (Fig. 5b), suggesting that $T_{iBAT}$ could be a driving force of $T_{core}$ changes. Further, the denervation of iBAT sympathetic nerves suppressed the hyperthermia induced by photoactivation (Fig. 5b), suggesting an essential function of the iBAT. Additionally, the expression of thermogenin uncoupling protein 1 (UCP1) in the iBAT was elevated 3-h post photoactivation (Fig. 5c). These results collectively show that iBAT thermogenesis is essential for the hyperthermic function of the LPB[Vglut2]→DMH pathway.

These increases in BAT thermogenesis prompted us to test whether long-term activation of this pathway would increase energy expenditure to lower body weight. Thus, we used a two-week photostimulation protocol to activate the LPB[Vglut2]→DMH projection in diet-induced obesity (DIO) mice fed with a high-fat diet (HFD) for 14 weeks (Fig. 5d).

We first confirmed that activating the LPB[Vglut2]→DMH projection four times a day can reliably induce an increase in body temperature (Fig. 5e). This indicates that the activation strategy is reliable and can effectively increase energy expenditure in mice. Although the photostimulation procedure itself appeared to curb weight gains in control mice, photoactivation of the LPB[Vglut2]→DMH projection further reduced body weight without affecting cumulative food intake (Fig. 5f–h). Consistently, photoactivation decreased the weights of iBAT and inguinal white adipose tissue (iWAT), reduced the adipose ratio, and increased EE during cold exposure (Fig. 5i–l). Therefore, chronic activation of this pathway increases EE and lowers body weight.

### Projection-specific transcriptomic analysis identifies DMH-projecting LPB[SST] neurons as cold-activated neurons

After showing that the LPB[Vglut2]→DMH pathway plays important roles in cold defense, we sought to identify neural types that comprise this pathway. We used a cell type- and projection-specific translating ribosome affinity purification sequencing technique (retroTRAP-seq) we developed[25], where retrograde AAVs carrying DIO-GFPL10 were injected into the DMH of Vglut2-Cre mice to express GFP-tagged ribosomal protein L10 in the LPB (Fig. 6a). The LPB was microdissected and GFP-tagged ribosomes were immuno-precipitated (IP) and used for subsequent sequencing. We analyzed the enrichment fold (IP/input) and found that 413 genes were upregulated (Fig. 6b and Supplementary Table 1). We compared our upregulated genes with PB-expressed genes obtained from the Allen Brain database and found several genes that overlapped, including *N4bp2os*, galanin (*Gal*), *SST*, and *Spint2* (Fig. 6c). Since a Cre strain was not available to label *n4bp2os*+ neurons, and *Gal* was not expressed in the LPB[25], we used SST-IRES-Cre to label *SST*+ neurons and performed further functional studies.

SST-Cre labeled neurons largely overlapped with SST immunoactivity (overlapped/SST+ = 89%; Supplementary Fig. 8a) and glutamate immunostaining[48] (overlapped/glu+ = 99%; Supplementary Fig. 8b), suggesting that this Cre line faithfully labeled *SST*+ neurons and that these neurons were glutamatergic. Quantifying the overlapping between LPB[SST] neurons and cold/warm-induced cFos suggested that LPB[SST] neurons accounted for only 8% of the neurons activated by warmth in the LPB (38°C, 2 h), but 20% of the neurons activated by cold (10°C, 2 h) (Fig. 6d, e). Also, more LPB[SST] neurons were activated in response to cold (~22%) than to warmth (~13%) (Fig. 6d, e). About 33% of DMH-projecting LPB[SST] neurons were sensitive to cold exposure, accounting for 16% of cold-activated LPB neurons (Fig. 6f). As a comparison, DMH-projecting LPB[Vglut2] neurons accounted for 38% of cold-activated LPB neurons (Fig. 1h, i). Using SST-Cre, we mapped the projection pattern of LPB[SST] neurons by injecting Cre-dependent ChR2 into the LPB (Fig. 6g, h and Supplementary Fig. 8c). As expected, LPB[SST] neurons projected to the DMH and other regions, including the MnPO, VMPO, and LPO (Fig. 6i and Supplementary Fig. 8d). Thus, our data indicate that glutamatergic LPB[SST] neurons exhibit a biased response to cold temperature, send projections to the DMH, and comprise up to ~40% (16%/38%) of cold-activated neurons within the LPB→DMH pathway.

### Monosynaptic connections between LPB[SST] and DMH neurons

To test whether there is a direct connection between LPB[SST] and DMH neurons, we expressed ChR2 in LPB[SST] neurons (Fig. 6h, i) and found that light stimuli induced EPSCs from DMH neurons (Fig. 6j). These currents were blocked by glutamate receptor antagonists (CNQX and AP5), and the latency after photoactivation was within the range of a monosynaptic connection (Fig. 6k, left two panels). Noticeably, only ~10% of randomly recorded DMH neurons exhibited EPSCs in response to light stimulation of SST+ terminals (Fig. 6k, right), making up ~1/3 of total LPB-innervating DMH neurons (Fig. 1k). To further verify this connection was monosynaptic, we first blocked EPSCs with tetrodotoxin (TTX) and then applied 4-aminopyridine (4-AP) to sensitize the postsynaptic current (Fig. 6l). Indeed, 4-AP restored EPSCs blocked by TTX, suggesting that LPB[SST] and DMH neurons form monosynaptic connections.

### The LPB[SST]→DMH pathway increases $T_{core}$ via iBAT thermogenesis and is required for cold defense

To investigate the function of the LPB[SST]→DMH pathway, we optogenetically activated LPB[SST & ChR2] terminals in the DMH (Fig. 7a, b). $T_{core}$ and physical activity were measured by telemetry probes and $T_{iBAT}$ was measured by infrared thermography. As expected, photoactivation of these terminals increased $T_{core}$ (~1.1°C) and $T_{iBAT}$, but not physical

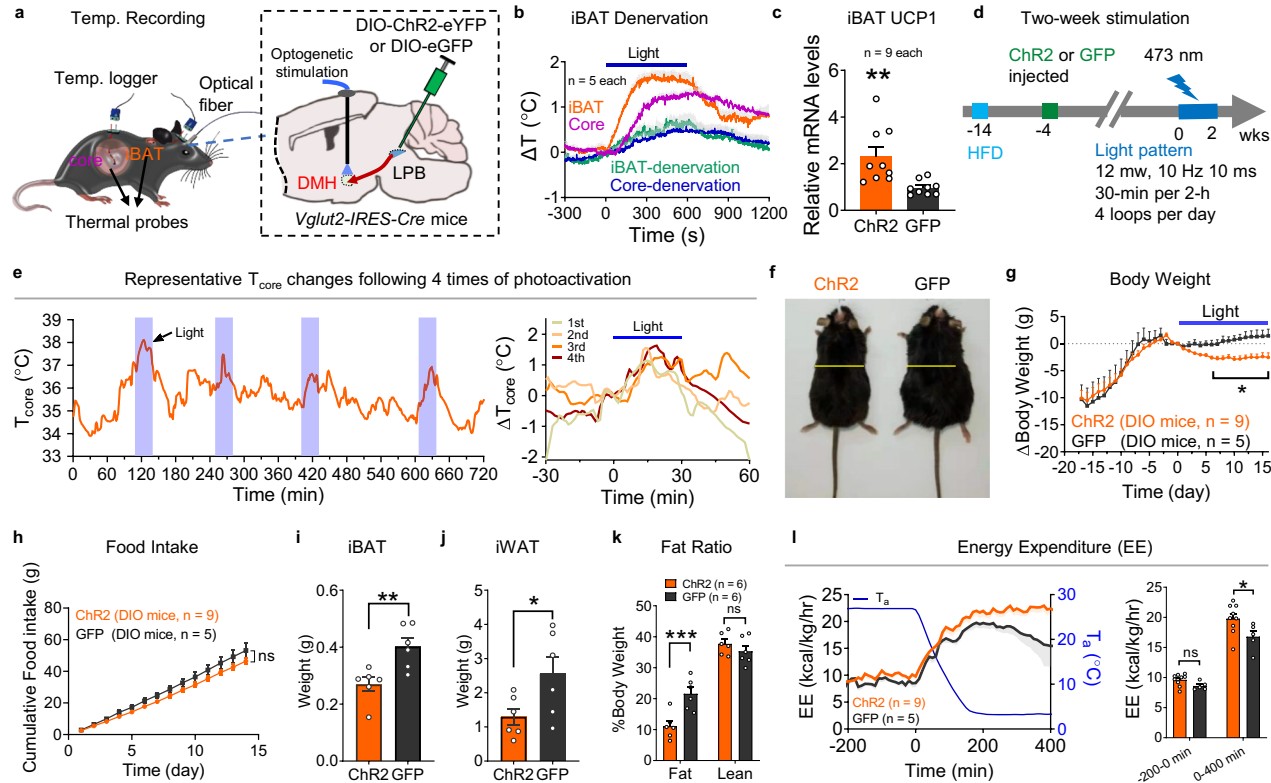

**Fig. 5 | Activation of the LPB$^{Vglut2}$ → DMH projection increases iBAT thermogenesis and suppresses body weight gain. a** Simultaneously recording $T_{core}$ and $T_{iBAT}$ in freely behaving mice at high temporal resolution using wired probes while activation of the LPB$^{Vglut2}$ → DMH projection. **b** Changes in $T_{core}$ and $T_{iBAT}$ in response to photoactivation of the LPB$^{Vglut2}$ → DMH projection in the sham group and iBAT denervation group (n = 5 mice each). Light pattern: 473 nm, 12 mW, 10 Hz, 10 ms, 2-s on 2-s off, 10 min. **c** Relative UCP1 expression levels in iBAT 3-h after photoactivation of the LPB$^{Vglut2}$ → DMH projection (n = 9 mice each). **d** Testing the effect of two-week photoactivation of the LPB$^{Vglut2}$ → DMH projection on body weight of DIO mice fed with HFD. **e** A representative trace of $T_{core}$ change (left panel) following four times of photoactivations of the LPB$^{Vglut2}$ → DMH projection in

a day and the comparison (right panel) of $T_{core}$ change for each photoactivation. **f** Mice after two weeks of photoactivation (left, ChR2 group; right, GFP group). **g, h** Changes in body weight (**g**) and cumulative food intake (**h**) during two weeks of photoactivation (ChR2, n = 9 mice; GFP, n = 5 mice). **i–l** Changes in iBAT weight (**i**), iWAT weight (**j**), body composition (**k**), and cold-induced energy expenditure (**l**) after two-week photoactivation (**i–k**, ChR2, n = 6 mice; GFP, n = 6; **l**, ChR2, n = 9 mice; GFP, n = 5 mice). All data are the mean ± sem, and (**g, h**) were analyzed by two-way RM ANOVA followed by Bonferroni's multiple comparisons tests, (**c**, and **i–l**) were analyzed by unpaired t-test. The *p*-values are calculated based on statistical tests in Supplementary Table 2. *$p ≤ 0.05$; **$p ≤ 0.01$; ns, not significant. Source data are provided as a Source data file.

activity (Fig. 7c–e). To test whether iBAT thermogenesis was required for this hyperthermia, we denervated iBAT sympathetic nerves and found it abolished the hyperthermic effect (Fig. 7f). In contrast to activation of the LPB$^{Vglut2}$ → DMH projection, activation of the LPB$^{SST}$ → DMH projection did not affect muscle shivering and heart rate (Fig. 7g, h). As a comparison, the POA-projecting LPB$^{SST}$ neurons in response to heat are twice as cold-responsive (Supplementary Fig. 9a, b), and activating the LPB$^{SST}$ → POA projection leads to hypothermia (Supplementary Fig. 9c, d). To fully rule out a contribution of POA projections from LPB$^{SST}$ neurons to hyperthermic responses, we used retrograde traveling AAVs carrying Cre-dependent FlpO to drive TeNT expression to block POA-projecting LPB$^{SST}$ neurons (Fig. 7i, j). Indeed, blocking POA-projecting LPB$^{SST}$ neurons did not affect the increases in $T_{core}$ and $T_{iBAT}$ induced by photoactivation of the LPB$^{SST}$ → DMH projection (Fig. 7k, l).

Furthermore, we modulated the activity of LPB$^{SST}$ somas via chemogenetics. Similar to terminal activation, activation of LPB$^{SST}$ somas via hM3D$_q$ caused increases in $T_{core}$ but not physical activity (Supplementary Fig. 9e–h). Chemogenetic activation also increased EE (Supplementary Fig. 9i). Together, we found that the LPB$^{SST}$ → DMH projection within the LPB → DMH pathway increases $T_{core}$ and EE via BAT thermogenesis selectively.

To test the necessity of the LPB$^{SST}$ → DMH pathway in thermoregulation, we blocked DMH-projecting LPB$^{SST}$ neurons via a

projection-specific lesion strategy that takes advantage of taCaspase3 (Fig. 7m). As expected, this lesion did not affect $T_{core}$ after warm exposure (35°C) (Fig. 7n). However, it impaired thermoregulation after cold exposure (4°C) (Fig. 7o). Together, these results demonstrate that the LPB$^{SST}$ → DMH pathway is required for cold defense by selectively controlling iBAT thermogenesis.

## LPB$^{SST}$ neurons target DMH$^{LepR}$ neurons to increase $T_{core}$

To determine the DMH cell types that were targeted by LPB$^{SST}$ neurons to increase $T_{core}$, we considered neurons expressing leptin receptor (LepR) and ChAT since they both regulate thermogenesis[36,38]. To block these two types of neurons while photoactivating LPB$^{SST}$ → DMH projections, we crossed SST-Cre with LepR-Cre or ChAT-Cre to obtain double Cre-positive mice, and then injected DIO-ChR2 and DIO-TeNT into the LPB and DMH, respectively (Fig. 7p, q). Although LepR-Cre and ChAT-Cre may also drive ChR2 expression in the LPB, LPB$^{LepR}$ neurons induce very mild hyperthermia[25] and LPB$^{ChAT}$ neurons barely project to the DMH (Allen Brain Atlas). Therefore, photoactivation of the LPB$^{SST+LepR/ChAT}$ → DMH projection would largely recapitulate the hyperthermia phenotype observed after photoactivating the LPB$^{SST}$ → DMH projection. Then, since SST-Cre is also expressed in the DMH (Fig. 7q, left panel), we photoactivated the LPB$^{SST}$ → DMH projection while blocking LepR$^+$ and SST$^+$, or ChAT$^+$ and SST$^+$

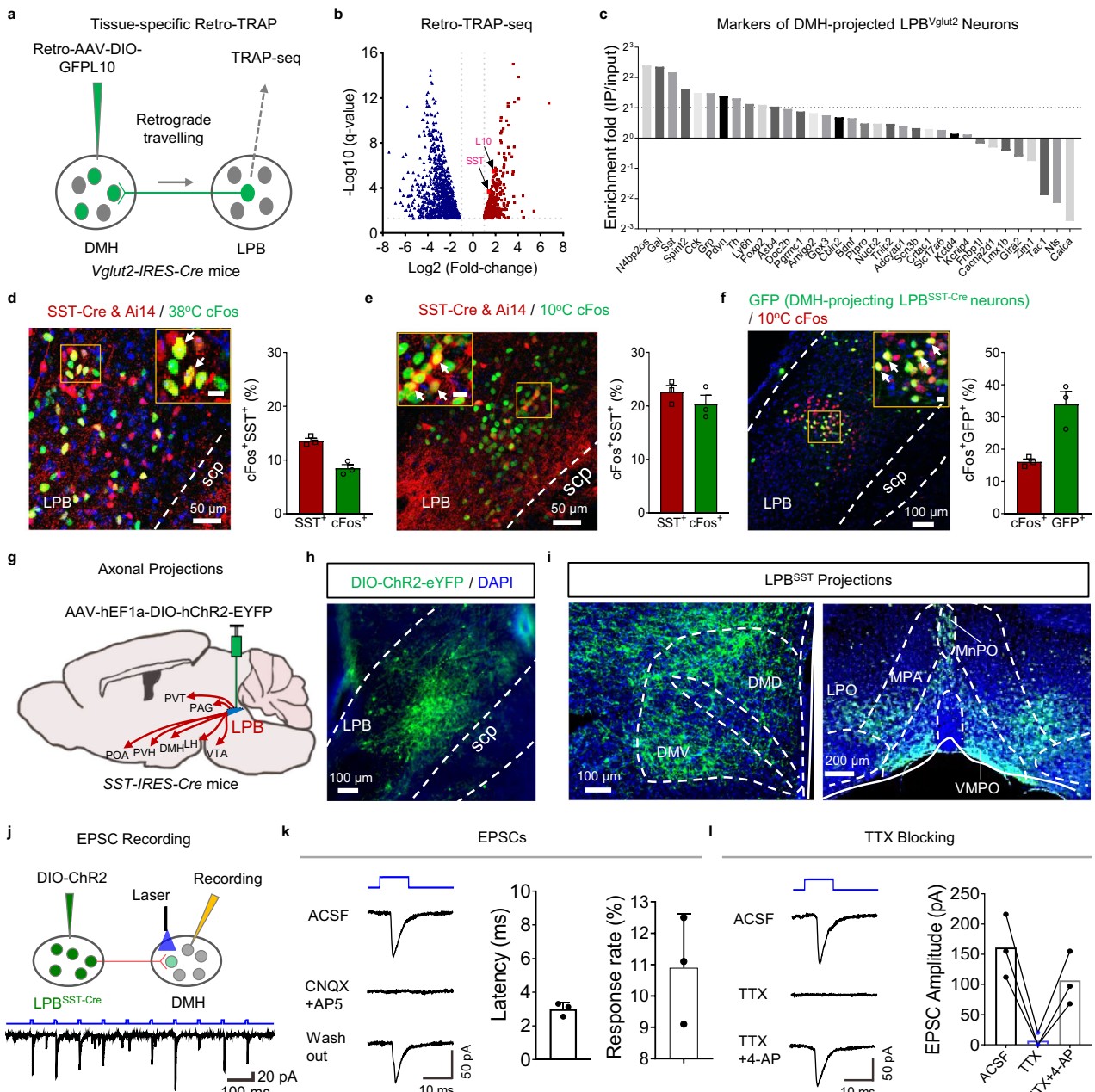

**Fig. 6 | Projection-specific transcriptomic analysis identifies DMH-projecting LPB^SST neurons as cold-activated neurons. a** Projection-specific transcriptomic analysis (retro-TRAP), where GFP-tagged translational ribosomes from DMH-projecting LPB^Vglut2 neurons were immunoprecipitated and associated mRNAs were sequenced. SST, somatostatin. **b** Volcano plots (q value versus log2 fold change) for LPB mRNAs after retro-TRAP sequencing. **c** Retro-TRAP fold enrichment (IP/Input) for PB-expressed genes downloaded from the Allen Institute. **d**, **e** Overlap between SST-IRES-Cre labeled neurons (Tdt⁺) in the LPB and the following immuno-positive neurons: warm-induced cFos (10°C, 2 h; **d**), and cold-induced cFos (38°C, 2 h; **e**). (n = 3 mice each). **f** Overlap between DMH-projecting LPB^SST neurons and cold-

induced cFos (10°C, 2 h) (n = 3 mice each). **g** Axonal tracing of LPB^SST neurons by ChR2, where AAV9-hEF1a-DIO-hChR2-EYFP was injected into the LPB of SST-IRES-Cre mice. PVH, paraventricular hypothalamic. **h**, **i** Expression of ChR2 in LPB^SST neural somas (**h**) and terminals in the POA or DMH (**i**). **j** Induction of EPSCs in DMH neurons by light stimulation of LPB^SST & ChR2 terminals (blue, 6 mW, 10 ms, 10 hz). **k** EPSCs were blocked by GluR antagonists D-AP5 and CNQX. Latency and response rate (3 of 28 randomly recorded neurons from 3 mice) are shown on the right. **l** EPSCs were blocked after TTX treatment but were restored by 4-AP treatment. All data are the mean ± sem. Scale bars in the enlarged panel of (**d**–**f**) are 10 μm. Source data are provided as a Source data file.

neurons in the DMH. Interestingly, blocking DMH^LepR+SST neurons blunted the induced hyperthermia, whereas blocking DMH^ChAT+SST neurons had no effect (Fig. 7r). Results from these blocking experiments are consistent with the function of DMH^LepR neurons, which increase BAT thermogenesis[36]. Therefore, we suggest that DMH^LepR neurons function as downstream targets of LPB^SST neurons to increase BAT thermogenesis.

## The raphe pallidus nucleus (RPa) functions downstream of the LPB → DMH pathway

In seeking to determine downstream targets of the LPB → DMH pathway, we considered the RPa since it is a synaptic target of DMH thermogenic neurons in both rats and mice[10,40,49]. We confirmed activation of the DMH^Vglut2 → RPa projection sufficiently increased T_core in mice by photoactivating the axon terminals of DMH^Vglut2 neurons in the RPa

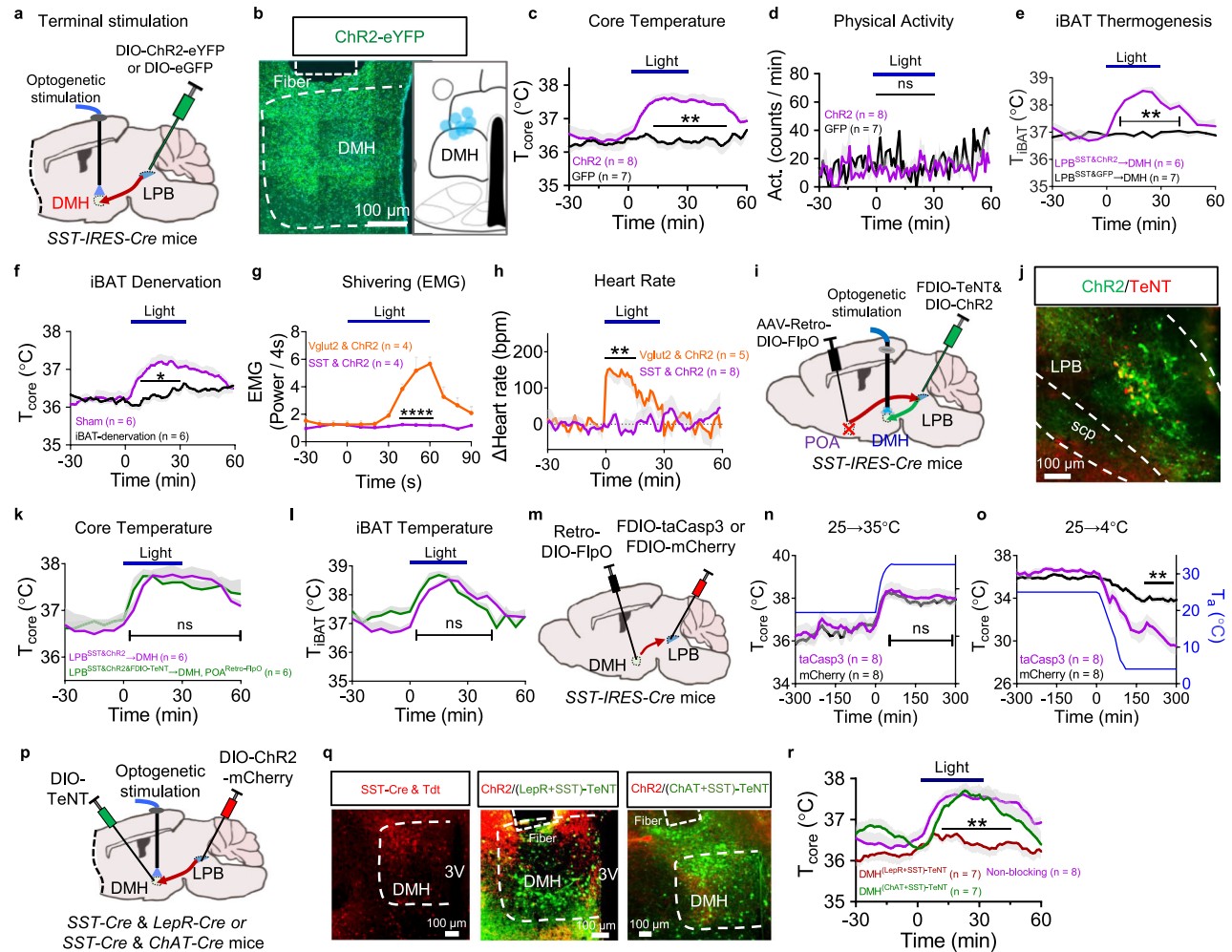

**Fig. 7 | The LPB$^{SST}$ → DMH projection is selectively required for iBAT thermogenesis during cold defense. a, b** Design to activate the LPB$^{SST}$ → DMH projection (**a**) and the representative expression of ChR2-eYFP (left) and summary of fiber tracts (shown as blue dots, right) in the DMH (**b**). This experiment was repeated at least 8 times independently with similar results. **c–e** Changes in T$_{core}$ (**c**), physical activity (**d**), and T$_{iBAT}$ (**e**) after photoactivation of the LPB$^{SST}$ → DMH projection. Animal numbers were indicated. **f** Denervation of the iBAT abolished the hyperthermia induced by photoactivation (*n* = 6 mice each). **g, h** Photoactivation of the LPB$^{SST}$ → DMH projection did not change nuchal muscle EMG (**g**; *n* = 4 mice each) and heart rate (**h**; SST, *n* = 8 mice; Vglut2, *n* = 5 mice). **i** Activating the LPB$^{SST}$ → DMH pathway while blocking POA-projecting LPB$^{SST}$ neurons using TeNT. **j** Representative expression of ChR2 (green) and TeNT (red) in the LPB. This experiment was repeated at least 6 times independently with similar results. **k, l** Changes in T$_{core}$ (**k**) and T$_{iBAT}$ (**l**) after photoactivation of LPB$^{SST}$ terminals in the

DMH while blocking POA-projecting LPB$^{SST}$ neurons (*n* = 6 mice each). **m** Deleting DMH-projecting LPB$^{SST}$ neurons by neural killing with taCasp3. **n, o** T$_{core}$ changes during warm (**n**) and cold (**o**) exposures after deleting DMH-projecting LPB$^{SST}$ neurons (*n* = 8 mice each). **p** Photoactivation of LPB$^{SST \& ChR2}$ terminals in the DMH after blocking DMH$^{LepR+SST}$ neurons or DMH$^{ChAT+SST}$ neurons. **q** Representative SST-Cre & Tdt (left panel) and LPB$^{SST \& ChR2}$ terminals (red) and TeNT (green) expression in the DMH (right two panels). This experiment was repeated at least 3 times independently with similar results. **r** Changes in T$_{core}$ after photoactivation of LPB$^{SST}$ terminal in the DMH while blocking DMH$^{LepR+SST}$ or DMH$^{ChAT+SST}$ neurons (*n* = 7 mice each). All data are the mean ± sem, and (**c–h, k, l, n, o, r**) were analyzed by two-way RM ANOVA followed by Bonferroni's multiple comparisons test. The *p*-values are calculated based on statistical tests in Supplementary Table 2. *$p \leq 0.05$; **$p \leq 0.01$; ****$p \leq 0.0001$; ns, not significant. Source data are provided as a Source data file.

(Supplementary Fig. 10a–c). Besides, RPa-projecting DMH Vglut2$^+$ and Brs3$^+$ neurons receive synaptic inputs from the LPB[35,40], suggesting the existence of an LPB → DMH→RPa pathway. To demonstrate the functional importance of this pathway, we photostimulated LPB$^{Vglut2 \& ChR2}$ terminals in the DMH while blocking RPa-projecting DMH neurons (Supplementary Fig. 10d). To do so, we infected LPB$^{Vglut2}$ somas with AAVs carrying DIO-ChR2, and injected a retrograde AAV carrying FlpO into the RPa to induce the expression of FlpO-dependent TeNT in the DMH (Supplementary Fig. 10d, e). The mCherry was used as a control for TeNT. Blocking RPa-projecting DMH neurons itself did not affect basal body temperature or physical activity compared with controls measured during the light cycle (Supplementary Fig. 10f). Next, we photoactivated LPB$^{Vglut2 \& ChR2}$ terminals in the DMH while blocking RPa-projecting DMH neurons. Strikingly, the hyperthermia induced by

photoactivation was abolished (Supplementary Fig. 10g), as well as autonomic cold defense responses associated with photoactivation, including physical activity, iBAT thermogenesis, and muscle shivering (Supplementary Fig. 10h–j). Hence, we show that RPa-projecting DMH neurons are required for the LPB → DMH pathway to mediate cold-defense function. We, therefore, suggest that the RPa is the primary downstream target of the LPB → DMH pathway to promote BAT thermogenesis and muscle shivering since the RPa is known to regulate these activities downstream of the DMH[10,35,40]. Yet, a concern was raised about whether physical activity changes were also mediated by the RPa since it has not been reported before.

We then tested whether the RPa plays a role in regulating physical activity by neural blocking (Supplementary Fig. 10k,l). Consistent with its role in thermogenesis, blocking RPa neurons via TeNT decreased

the basal $T_{core}$ during the dark cycle compared with GFP controls (Supplementary Fig. 10m). As expected, the hyperthermia induced by photoactivation of LPB$^{Vglut2\ \&\ ChR2}$ neural terminals in the DMH was abolished after blocking RPa neurons (Supplementary Fig. 10n), further supporting that the RPa is the downstream target of the LPB → DMH pathway. To our surprise, blocking the RPa itself caused a compulsive-like circling behavior in the cage (Supplementary Fig. 10o), which resulted in hyperactivity and prevented us from judging its role in regulating physical activity during cold defense. Therefore, the roles of the RPa in regulating physical activity require further studies.

## Discussion

Hypothermia caused by cold or starvation is a major threat to life[4,5,50]. Therefore, the ability to prevent hypothermia under normal conditions and to rapidly recover from a bout of hypothermia is critical for animal survival and fitness. Here, we found that a previously uncharacterized pathway in mice, namely the LPB → DMH pathway, functions in parallel to the LPB → POA pathway to rapidly increase thermogenesis during cold defense and quickly recover from hypothermia caused by cold exposure in mice (Fig. 4). This parallel-circuit model (illustrated in Fig. 8) deepens our understanding of how neuronal circuits may affect thermoregulation and may represent a basic network in regulating other essential homeostatic behaviors.

The LPB → POA pathway has been suggested to be important for cold defense in rats. Pharmacological activation of LPBel neurons with N-methyl-D-aspartic acid (NMDA) increases BAT thermogenesis and HR in rats, and these phenotypes are suppressed by antagonizing glutamate receptors in the MnPO[39]. Consistent with this, our data

suggest that LPB neurons mainly innervate the MnPO, VMPO, and LPO in mice (Fig. 1b). However, recent optogenetic manipulations in mice failed to verify a cold-defense function of the LPB → POA pathway, whose activation caused hypothermia alone[25,26]. These conflicting results raise the issue of whether there are species-specific roles played by the LPB → POA pathway in cold defense. To address this issue in mice, we mapped the temperature sensitivity of the LPB → POA/DMH pathways via cFos staining. As expected, warm-activated neurons predominantly reside in the LPB → POA pathway. In contrast, cold-activated neurons are equally distributed between the two pathways (Fig. 1e–h). Blocking either of these two pathways impaired cold defense to a similar extent (Fig. 3), suggesting both pathways are required for physiological defense responses to cool/cold temperatures. The reason for not seeing a hyperthermia phenotype after bulk optogenetic activation of the LPB → POA pathway might be due to a masking effect of heat-defense neurons within this pathway[25,26]. Thus, more selective genetic manipulations are needed to reveal the identity of cold-defense neurons within the LPB → POA pathway. Most notably, blocking both LPB → POA/DMH pathways exhibited a much stronger impairment in cold defense (Fig. 3h–k), suggesting a cumulative or additive effect of the two pathways. Thus, these two pathways assemble into a parallel circuit. Also, it is noteworthy that the LPB neurons projecting to the POA/DMH region also have projections to other brain regions involved in thermoregulation, such as the LH and VMH (Supplementary Fig. 3). However, the LPB → LH projection was not found to be necessary for cold-induced thermogenesis[51] and VMH-projecting LPB neurons were not responsive to cold stimuli[52,53]. Further investigation is necessary to ascertain the existence of parallel

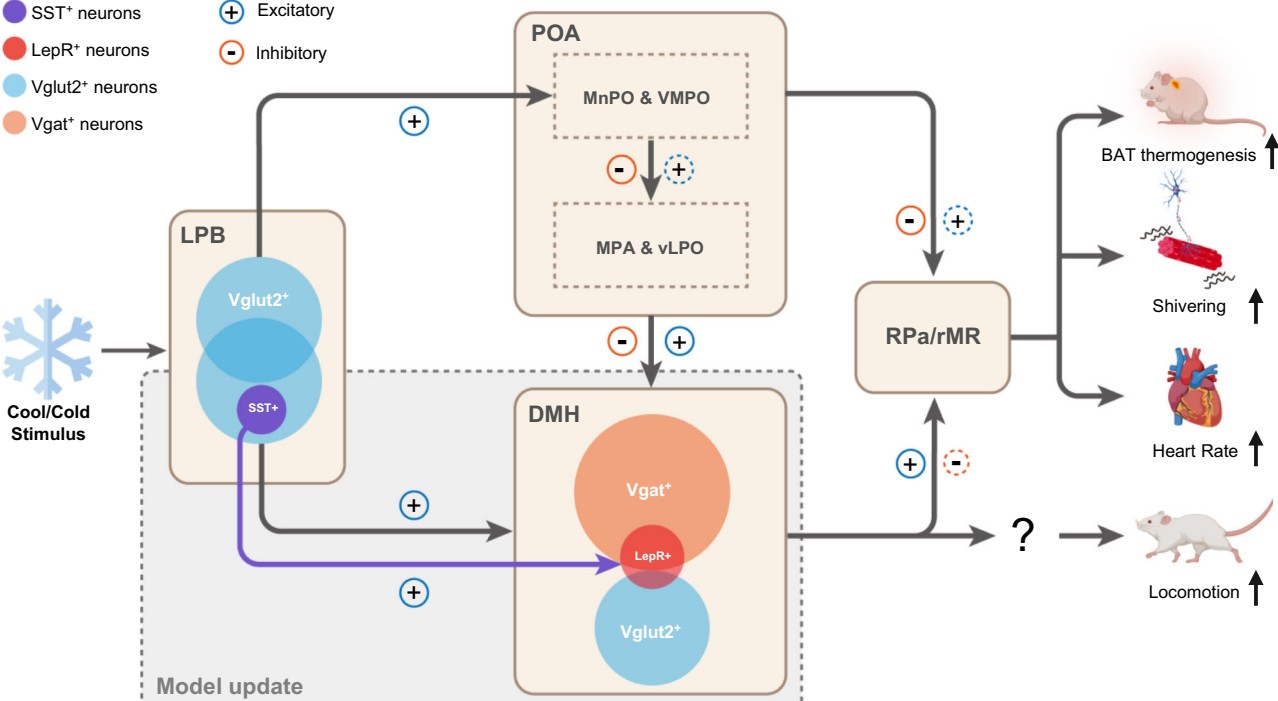

**Fig. 8 | Proposed parallel circuits for cold defense.** The LPB → POA → DMH/RPa circuitry has been proposed before[10] and the model updated by this study is shaded in grey color. Afferent cool/cold signals activate Vglut2$^+$ neurons in the LPB. Among these cold-activated neurons (cFos$^+$; 100%), 45% projected to the POA (mainly the MnPO, VMPO, and LPO), 38% projected to the DMH, and 20% projected to both regions. Of note, this percentage might be underestimated due to the limited retrograde efficiency. Projection-specific neural blockings suggest that both the LPB → POA and LPB → DMH pathways are required in cold defense, where the two pathways contribute equivalently and cumulatively to cold defense and therefore form a parallel circuit. Within the LPB → DMH pathway, 60% of LPB-innervated DMH

neurons are Vgat$^+$, while 20% of them are Vglut2$^+$. Activation of the LPB → DMH pathway induces strong cold-defense responses, including increases in BAT thermogenesis, muscle shivering, heart rate, and locomotion. Both the DMH$^{Vgat}$ and DMH$^{Vglut2}$ neurons are required to support the cold-defense function of the LPB → DMH pathway. Additionally, a subpopulation of SST$^+$ neurons in the LPB targets DMH$^{LepR}$ neurons to promote BAT thermogenesis selectively, suggesting a genetically defined neural projection controls specific cold-defense activities. Downstream of the DMH, the RPa or rostral medullary raphe region (rMR) is known to regulate BAT thermogenesis, muscle shivering, and heart rate, while the regulation of locomotion is not clear. This schematic was created with BioRender.com.

pathways that operate in conjunction with the LPB→POA/DMH projections.

Interestingly, this LPB→DMH pathway may also function in rats since cold exposure also induces cFos expression in DMH-projecting LPB neurons in rats[39]. Hence, mice and rats may use an evolutionarily conserved neural circuit that we describe here, namely the parallel circuit comprised of the LPB→POA/DMH pathways, to control cold defense. The evolution of parallel neural circuits in cold defense not only enables resilience to hypothermia but also provides a scalable, robust, and efficient network in heat production when both pathways are recruited.

Although the LPB is considered the 'relay station' for transmitting warm and cold signals from the spinal cord to the POA. Recent findings in mice reveal that warm signals are relayed by different types of neurons to control different warm defense activities[25,26]. For instance, we have shown that warm-activated Prodynorphin+ (Pdyn+) and cholecystokinin+ (CCK+) neurons in the LPB are responsible for inhibiting BAT thermogenesis and promoting skin vasodilation, respectively[25]. Here, we show that cold signals were relayed by a parallel circuit, namely the LPB→POA/DMH pathways, to control a variety of cold defense activities (Figs. 3 and 4). Among these activities, we showed that the LPB$^{SST}$→DMH$^{LepR}$ pathway governs iBAT thermogenesis, suggesting a genetically defined projection controls specific cold-defense activities. We reasonably speculate that other cold defense activities, including heart rate and muscle shivering, are also controlled by genetically defined neural projections. Together, these findings suggest that the LPB is a critical thermoregulation center that relays thermal afferent signals via different genetically defined neurons or projections as part of a feed-forward mechanism to control different thermal effector activities.

Whether LPB neurons simply relay these afferent signals or actively process them remains an interesting question. Considering that the LPB receives diverse inputs from brain regions with functions related to internal state sensing and fear, including the nucleus of the solitary tract (NTS)[25,54], the rostral ventrolateral medulla[55], and the paraventricular hypothalamus[56], we propose that the LPB may also incorporate other types of signals to modify thermal afferent signals. In support of this hypothesis, it has recently been suggested that the LPBel receives input from the NTS and sends projections to the posterior subthalamic nucleus to mediate innate fear-associated hypothermia[54].

## Methods

### Animals
Animal care and use conformed to institutional guidelines of ShanghaiTech University, Shanghai Biomodel Organism Co., and governmental regulations. All experiments were performed on male adult mice (8−16 weeks old). Mice were housed under controlled temperature (22−25 °C) and humidity (50-70%) unless specified in a 12-hour reverse light/dark cycle (light time, 9 pm to 9 am) with food and water ad libitum. Mice were fed either a high-fat diet (HFD, Research Diets, #D12492) or regular chow food (SLAC, #M03) as described in the text and figure legends. The following strains were used: Vgat-IRES-Cre (JAX stock no.028862), Vgat-T2A-FlpO (JAX stock no.029591), Vglut2-IRES-Cre (JAX stock no. 028863), SST-IRES-Cre (JAX stock no. 028864), LepR-Cre (JAX stock no. 008320), ChAT-Cre (JAX stock no. 006410), C57BL/6 J (JAX stock no. 000664), Ai14 (JAX stock no. 007908).

### Stereotaxic brain surgeries and viral injection
Mice were given general anesthesia with isoflurane during stereotaxic injection. The injection was performed using a small animal stereotaxic instrument (David Kopf Instruments, #PF-3983; RWD Life Science, #68030; Thinker Tech Nanjing Biotech, #SH01A). AAV virus ( - 0.15 μl,

unless specified) was delivered through a pulled-glass pipette and a pressure micro-injector (Nanoject II, #3-000-205 A, Drummond) at a slow rate (23 nl min$^{-1}$) with customized controllers. The coordinates of viral injection sites include the LPB (AP, −4.95 mm; ML, ± 1.5 mm; DV, −3.6 mm), the DMH (AP, −1.25 mm; ML, ± 0.3 mm; DV, −5.2 mm), the VMPO (AP, 0.75 mm; ML, 0 mm; DV, −5.0 mm), the vLPO (AP, 0.35 mm; ML, ± 0.75 mm; DV, −5.5 mm) and the RPa (AP, −5.8 mm; ML, 0 mm; DV, −5.7 mm). The injection needle was withdrawn 10 min after the end of the injection. During surgeries, a feedback heater was used to maintain core body temperature at 36 ± 1 °C. The optical fiber (200 μm in diameter, Inper Inc., China) was chronically implanted in the LPB (AP, −4.95 mm; ML, ± 1.5 mm; DV, −3.4 mm), and the DMH (AP, −1.25 mm; ML, ± 0.3 mm; DV, −4.9 mm) and secured with dental cement (C&B Metabond®, Parkell, Japan). The bregma sites and anatomy are indicated directly on relevant figures. Mice were transferred to housing cages for 3-4 weeks before performing behavioral evaluations. After behavioral tests were finished, mice were perfused to check the virus expression and fiber insertion. Data from mice that showed little or no viral expression or had a fiber insertion that missed the target (often 0-20%) were excluded from the analysis. We used a mouse brain atlas (Franklin and Paxinos, 2008, 4$^{th}$ edition) to determine coordinates for injection sites. To construct the heatmap in Supplementary Figs. 5 and 7, we converted the photomicrograph of the injection site into a binary one, applied a Gaussian filter to remove the remaining noise, and stacked them together using ImageJ (version 1.8.0). The following viruses were used: AAV2/9-hSyn-DIO-EGFP-WPRE-hGHpA, AAV2/8-hSyn-DIO-mCherry-WPRE-Pa, were purchased from Shanghai Sunbio Medical Biotechnology Co. AAV2/9-EF1α-DIO-hM4D(Gi)-mCherry was purchased from BrainVTA (Wuhan) Co.,Ltd, AAV2-Retro-hSyn-DIO-mCherry-WPRE-pA, AAV2/9-hSyn-DIO-GCaMP6s-WPRE-SV40pA, AAV2-Retro-hSyn-DIO-GCaMP6s-WPRE-SV40pA, AAV2/9-hSyn-DIO-jGCaMP7b-WPRE-pA, AAV2/9-hEF1a-DIO-hChR2(H134R)-mCherry, AAV2/9-hSyn-DIO-hM3Dq-EGFP, AAV2/9-hEF1a-DIO-hChR2(H134R)-EYFP, AAV2/9-hEF1a-FDIO-hChR2(H134R)-EYFP, AAV2/9-hEF1a-DIO-mCherry-P2A-TeNT-WPRE-pA, AAV2-Retro-hEF1a-DIO-GFPL10-WPRE-hGHpA, AAV2/9-hEF1a-FDIO-taCasp3-TEVp, AAV2/8-hEF1a-FDIO-mCherry, scAAV2/1-hSyn-FlpO-pA, AAV2/8-hEF1a-FDIO-hM3Dq-mCherry, scAAV2/1-hSyn-Cre-pA, AAV2/5-hSyn-DIO-GFP-P2A-TeNT, AAV2/9-hEF1a-FDIO-mCherry-2A-TeNT-WPRE-pA, AAV2/9-hSyn-eGFP-2A-TeNT-WPRE-hGHpA, AAV2/9-hSyn-MCS-EGFP-3FLAG, AAV2-Retro-CAG-DIO-FlpO, AAV2/9-hSyn-Cre-off-FlpO-on-EGFP-T2A-TeNT, AAV2/9-hSyn-Cre-off-FlpO-on-EGFP were purchased from Shanghai Taitool Bioscience Co. All were used with titers about $2 - 5 \times 10^{12}$.

### Immunohistochemistry
Mice were anesthetized with isoflurane and perfused transcardially with PBS and 4% paraformaldehyde (PFA) in PBS. Brain tissues were post-fixed overnight at 4 °C, then sectioned at 50 μm using a vibratome (Leica, VT1200S). Brain slices were collected and blocked with blocking solution (Roche, #11096176001) for 2-h at room temperature and subsequently incubated with primary antibodies (1:1000, unless specified) for 2 days at 4 °C. Then, the samples were washed three times in PBST (PBS with 0.1% Triton X-100, v/v) before incubating in secondary antibodies (1:1000, unless specified) overnight at 4 °C. For glutamate, GABA, and somatostatin staining, mice were perfused transcardially with PBS followed by 50 ml 4% PFA without post-fixation. Brains were dehydrated in 20% sucrose for 1 day and 30% sucrose for 2 days at 4 °C, then sectioned at 40 μm thicknesses on a cryostat microtome (Leica, CM3050s). Brain slices were collected and blocked with the blocking solution containing 10% normal goat serum (v/v) and 0.1% Triton X-100 (v/v) in PBS overnight at 4 °C and subsequently incubated with primary antibodies (1:500) for 8-h at room temperature. After that, the slices were washed three times in PBST (PBS with 0.7% Triton X-100, v/v) before being incubated in secondary antibodies (1:1000) for 2-h at room temperature. Brain sections were

washed three times in PBST and cover-slipped with DAPI Fluoromount-G mounting medium (SouthernBiotech, #0100-20). Images were captured on a Nikon A1R or Leica SP8 confocal microscope or Olympus VS120 Virtual Microscopy Slide Scanning System. The following antibodies were used: Chicken anti-GFP(Abcam, #ab13970), Guinea pig anti-cFos (Synaptic systems, #226004), Rabbit anti-cFos(Synaptic systems, #226003), Rat anti-RFP (Chromotek, #5F8), Rabbit anti-Glutamate (Sigma, #G6642), Rabbit anti-GABA (Sigma, #A2052), Rabbit anti-Somatostatin(ImmunoStar, #20067), DyLight 488 conjugated goat anti-chicken (Invitrogen,#SA5-10070), Alexa Fluor 594 conjugated goat anti-rat IgG (Invitrogen, #A-11007), Alexa Fluor 594 conjugated goat anti-Guinea pig IgG (Invitrogen, #A-11076), Alexa Fluor 594 conjugated goat anti-rabbit IgG(Jackson, #111-585-144), Alexa Fluor 488 conjugated goat anti-rabbit IgG(Jackson,#111-545-003), Alexa Fluor 647 conjugated goat anti-rabbit(Invitrogen,#A21244).

### Triple-labeling of warm/cold-activated cFos, POA- and DMH-projecting LPB[Vglut2] neurons

Vglut2-IRES-Cre mice were injected with AAV-retro-DIO-GFPL10 into the POA and AAV-retro-DIO-mCherry into the DMH. Four weeks following the injection, mice were exposed to a warm (38°C) or cold (10°C) environment in an incubator for two hours, then were perfused with PBS and 4% PFA and processed following the immunohistochemistry protocol described above. POA-projecting LPB[Vglut2] neurons were labeled with GFP. DMH-projecting LPB[Vglut2] neurons were labeled with mCherry. Warm- or cold-activated cFos in the LPB neurons were labeled by cFos immunostaining (rabbit anti-cFos, Synaptic systems, #226003, 1:10000) colored with Fluor 647-conjugated secondary antibody (Alexa Fluor 647 conjugated goat anti-rabbit, Invitrogen, #A21244, 1:1000). Images were captured on a Nikon A1R confocal microscope.

### Slice physiological recording

Brain slice recording and data analysis were performed similarly as described in[14,25]. Briefly, slices containing the DMH regions were prepared from adult mice anesthetized with isoflurane before decapitation. Brains were removed and placed in ice-cold oxygenated (95% $O_2$ and 5% $CO_2$) cutting solution (228 mM sucrose, 11 mM glucose, 26 mM $NaHCO_3$, 1 mM $NaH_2PO_4$, 2.5 mM KCl, 7 mM $MgSO_4$, and 0.5 mM $CaCl_2$). Coronal brain slices (250 μm) were cut using a vibratome (VT 1200 S, Leica Microsystems, Germany). The slices were incubated at 32 °C in oxygenated artificial cerebrospinal fluid (ACSF: 119 mM NaCl, 2.5 mM KCl, 1 mM $NaH_2PO_4$, 1.3 mM $MgSO_4$, 26 mM $NaHCO_3$, 10 mM glucose, and 2.5 mM $CaCl_2$) for 1 h, and were then kept at room temperature under the same conditions before transfer to the recording chamber. The ACSF was perfused at 2 ml/min. The acute brain slices were visualized with a 40x Olympus water immersion lens, differential interference contrast optics (Olympus Inc., Japan), and a CCD camera (IR1000, Dage-MTI, USA).

Patch pipettes were pulled from borosilicate glass capillary tubes (#BF150-150-86-10, Sutter Instruments, USA) using a P-97 pipette puller (Sutter Instruments, USA). For EPSC recordings, pipettes were filled with solution (in mM: 130 CsMeSO$_3$, 1 $MgCl_2$, 1 $CaCl_2$, 10 HEPES, 2 QX-314, 11 EGTA, 2 Mg-ATP, and 0.3 Na-GTP; pH 7.3; 295 mOsm). Cells were clamped at −70 mV. To block EPSCs, 10 μM CNQX (Sigma) and 50 μM DL-AP5 (Tocris) were applied to the bath solution. For IPSC recordings, cells were clamped at 0 mV. 10 μM CNQX (Sigma) and 50 μM DL-AP5 (Sigma) were applied to the bath solution. To block IPSCs, 25 μM bicuculline (Tocris) was applied to the bath solution. The resistance of pipettes varied between 3–5 MΩ. The signals were recorded with MultiClamp 700B, Digidata 1440 A interface, and Clampex 10 data acquisition software (Molecular Devices). After the establishment of the whole-cell configuration, series resistance was measured. Recordings with series resistances of > 20 MΩ were rejected. Blue light pulses were delivered through the 40X objective of the

microscope with the X-Cite LED light source. The light power density was adjusted to 7 mW/mm$^2$.

### Calcium fiber photometry

Following injection of an AAV2/9-hSyn-DIO-GCaMP6s or AAV2-Retro-hSyn-DIO-GCaMP6s (Shanghai Taitool Bioscience Co.) viral vector, an optical fiber (200 μm O.D., 0.37 numerical aperture, Inper Inc., China) was placed 150 μm above the viral injection site. Post-surgery mice were transferred to housing cages for at least three weeks before any experiment. Fluorescence signals were captured with a dual-channel fiber photometry system (Fscope, Biolinkoptics, China) equipped with a 488-nm excitation laser (OBIS, Coherent), 505-544-nm emission filter, and a photomultiplier tube (Hamamatsu, #R3896). The gain (voltage) on PMT was set to 600 V. The laser power at the tip of the optical fiber was coordinated to 25 − 40 μW to minimize bleaching. The analog voltage signals were low-pass filtered at 30 Hz, digitalized at 100 Hz, and then acquired by Fscope software (Biolinkoptics, China).

We also used a fiber photometry system from Inper Ltd. to record the fluorescence signals from GCaMP6s. Light from a 470-nm LED was bandpass filtered, collimated, reflected by dichroic mirrors, focused by a X20 objective, and delivered at a power of 25–40 μW on the tip of the fiber optic cannula. Emitted fluorescence from GCaMP6s was bandpass filtered and focused on the sensor of a CMOS camera. The end of the fiber was imaged at a frame rate of 60 fps with the Inper Studio, and the mean value of the ROI of an end-face of the fiber was calculated using Inper Analysis software. To serve as an isosbestic control channel, 410-nm LED light was delivered alternately with 470-nm LED light.

To sterilize the surface, the optical fiber and the distal ends of the cord were cleaned using 75% ethanol. All animals were allowed to acclimate for at least 30 min after the fiber cord attachment. To control cage floor temperature changes, we used a Peltier controller (#5R7-001, Oven Industry) with a customized Labview code (National Instrument) to control Peltier floor plates (15 × 15 cm) for each mouse.

### Fiber photometry data analysis

To calculate the fluorescence change ratios, we analyzed the raw data using Fscope software or Inper Analysis software with customized MATLAB code. We segmented the data based on behavioral events within individual trials. The values of fluorescence change ($\Delta F/F_0$) were derived by calculating $(F − F_0)/F_0$, where $F_0$ is the baseline fluorescence signal averaged in a 120 s time window prior to temperature changes. The peak $\Delta F/F_0$ in (Fig. 2e) represents the maximum $\Delta F/F_0$ after warming or cooling the floor. The cooling phase of $T_{floor}$ in (Fig. 2f) is defined as the period that $T_{floor}$ changes from the starting temperature to a target temperature, and the steady phase is defined as the period that $T_{floor}$ maintains at the target temperature. Finally, we plotted the average fluorescence changes ($\Delta F/F_0$) with different events using MATLAB and GraphPad Prism 8 (GraphPad).

### Cold plate test

For quantification of spontaneous pain, the cold plate test was performed by using the test chamber with a homemade thermostatic plate, which is able to reach variable temperatures. Mice were placed on the plate fixed at 4 °C, free to move and walk. Spontaneous nociceptive behavior was quantified by counting the number of paw lifts recorded in a trial of 5 minutes. We also measured the limb withdrawal latency, which defines the latency of the mice's first limb lift after the mice had been placed on the cold plate.

### Temperature preference test

Mice were individually housed at least seven days before testing. Mice with the blocking of LPB-innervating POA/DMH neurons (Supplementary Fig. 5j) were placed in a chamber containing two identical

adjacent plates with one set to $30 \pm 0.5\,°C$ and the other adjusted to various temperatures as indicated (30, 16, 10, 6, and $35\,°C$). Mice were free to explore for five minutes, and each test was videotaped for later analysis. The mice moving trajectories were analyzed by Digbehv video-recording software (Jiliang, Shanghai, China) to calculate the total time spent on each side. All tests were performed in the dark phase between 9 am and 6 pm.

## Metabolic measurement

For DREADDs and TeNT mice, energy expenditure, locomotor activity, and core body temperature were monitored by the Comprehensive Lab Animal Monitoring System with Temperature Telemetry Transmitter (CLAMS; Columbus Instruments, with G2 E-Mitter transponders) and ambient temperature was labeled in related figures. The data was acquired at a 10-min interval, as shown in the figures. Temperature transponders were implanted into the peritoneal cavity 3–5 days before testing. Mice were adapted in the chambers for 2 days before giving saline (volume (µl) = 10 x body weight (grams)), and CNO (ENZO, #BML-NS105-0025, I.P., 2.5 mg/kg body weight). Stimuli (drugs or temperature) were delivered in the dark phase.

## Core body temperature, heart rate, physical activity, and iBAT temperature measurement

The core temperature, heart rate, and physical activity were recorded at a 1-min interval by VitalView Data Acquisition System Series 4000 (Starr Life Sciences Corp., Oakmont, USA) unless specified. To record core temperature and physical activity, we used G2 E-Mitter transponders implanted intraperitoneally. To record core temperature, heart rate, and physical activity, we used the G2-HR E-Mitter transponders. The HR E-Mitter was slipped into the abdominal cavity along the sagittal plane, and both leads were brought out of the abdominal incision. We then secured the negative lead (black) against a chest muscle on the animal's right. We secured the positive lead (shorter - clear) against a chest muscle on the animal's left. After recovery, mice were placed in their homecages on top of the ER-4000 Receivers. The heart rate is then recorded by the VitalView system as a beats-per-minute value based on computation from the R-R (R wave of the QRS complex) interval. Temperature challenges, laser stimulations and drugs were delivered in the dark phase.

The interscapular brown adipose tissue (iBAT) temperature ($T_{iBAT}$) was measured using a thermal infrared camera (A655sc, FLIR). And we analyzed the infrared images using the FLIR Tools software (Teledyne FLIR). To measure the iBAT temperature, the hair on top of the iBAT was shaved 3-5 days before measurement. The $T_{iBAT}$ was evaluated by measuring the average temperature in the interscapular region. We used the factory settings to convert raw pixel counts to units of temperature and analyzed the pictures taken from the same angle.

For wired recording with high temporal resolution shown in Fig. 5a, b, we applied a modified thermocouple unit as we reported before[25]. Two T-type thermocouples (TT-40, Omega) were implanted in the abdomen and within the iBAT to measure the core and iBAT temperatures, respectively. Mice were recovered for 3 – 5 days after surgery. To ensure these probes were accurate, we calibrated them using water baths and a standard thermal probe. The NI data acquisition card (USB-TC01, National Instrument) and supporting software were used for data collection, with a 1-Hz sampling rate.

## Heat production calculation

The normalized heat production of mice after photoactivation of LPB$^{Vglut2\ \&\ ChR2}$ terminals in the DMH under different ambient temperatures, as shown in Fig. 4r. The heat production was calculated according to the paper[57]. The original equation is: heat loss rate (HLS) = $Cp_{mouse}*(T_{core2}-T_{core1})/(t_2-t_1)/[(T_{core1}+T_{core2})/2-(T_{a1} + T_{a2})/2]$, which $Cp_{mouse}$ represents the heat production capacity; in another

form: $Cp_{mouse} = [(T_{core1}+T_{core2})/2-(T_{a1} + T_{a2})/2] * HLS * (t_2-t_1) / (T_{core2}-T_{core1})$. While in our situation: $t_2 = 30$ min, $t_1 = 0$ min, and $T_{a1} = T_{a2}$, thus, the equation can be simplified as $Cp_{mouse} = 30 * HLS * ((T_{core\_mean} -T_a) / \Delta T_{core})$, and the parameter – HLS relatively constant; so, the heat production capacity somehow correlated with the rate of $(T_{core\_mean}-T_a) / \Delta T_{core}$. Then the heat production of each $T_a$ was normalized by dividing the mean value of the heat production of $T_a = 30\,°C$.

## Electromyogram (EMG) recording

Mice were anesthetized with isoflurane and fixed with the animal stereotaxic instrument to avoid movement. An aluminum thermal pad with circulating water was used to maintain mice's body temperature at $36 \pm 1.5\,°C$. The rectal temperature was monitored by a thermocouple to reflect the body temperature under anesthesia. Another thermocouple to record skin temperature was taped onto the abdominal skin. Temperatures were recorded by NI data acquisition card and supporting software (National Instrument). Handmade electrodes for EMG recording were inserted into nuchal muscles after scissoring the skin, and the signal was amplified (×1000) and filtered (10 – 1000 Hz) with an 1800 2-Channel Microelectrode AC Amplifier (A-M Systems, Sequim, WA, USA). The laser stimulation pattern was 6 mW, 10 Hz, 10 ms on, for 60 s or 120 s as shown in the figures. The EMG amplitude was quantified (Spike 2, CED, Cambridge, UK) in sequential 4-sec bins as the square root of the total power (root mean square) in the 0 – 500 Hz band of the auto spectra of each 4-s segment[58].

## iBAT sympathetic nerve denervation

Mice were anesthetized with isoflurane. The hair in the targeted area was shaved, and the area was sterilized with 95% ethanol-soaked sterile gauze. A midline incision was made in the upper dorsal skin to expose both iBAT pads, and styptic powder (KELC, USA) was applied to the area to prevent bleeding. We gently exposed the medial, ventral surface of both pads to visualize nerves beneath the pads. All five nerves innervating both pads were cut off a length of 2-3 mm to avoid nerve regeneration under a stereoscope. Mice were housed in the thermoneutral environment for 5-7 days before tests.

## Quantitative (or real-time) PCR

RNA was extracted using TRIzol reagent (Sangon, #B511311-0100). The RNA quality and quantity were determined using a NanoDrop 5500 (Thermo). mRNA was reverse transcribed using a High-Capacity cDNA Reverse Transcription Kit (Takara, #RR047A) according to the manufacturers' instructions and processed for quantitative real-time PCR using the SYBR Green PCR system (Takara, #RR820A). The primers used to amplify a fragment of Ucp1 were 5'-ACTGCCACACCTCCAGTCATT-3' and 5'- CTTTGCCTCACTCAGGATTGG-3'. Mouse β-actin was used as the endogenous control to which sample values were normalized. The sequences used to amplify a fragment of β-actin were 5'- GTGACGTT-GACATCCGTAAAGA −3' and 5'-GCCGGACTCATCGTACTCC −3'. Expression levels were calculated based on the $2^{-\Delta\Delta Ct}$ method.

## Food intake, body weight, and body composition analysis

For the optogenetic experiments shown in Fig. 5d–k, the body weight and food intake were measured in the home cage. For DIO mice, the 5-week-old mice were fed with an ad libitum high-fat diet (HFD, Research Diets, #D12492) to drive body weight gain. AAVs carrying Cre-dependent GFP control or ChR2-tdTomato were injected in the DMH of Vglut2-IRES-Cre mice ten weeks after HFD feeding. Mice's body weight and food intake were measured daily ( ~ 4 pm). Four weeks after viral injections, mice were photo-stimulated to activate the LPB$^{Vglut2}$ terminals in the DMH for two weeks, as indicated in the figure.

After two-week photoactivation of the LPB$^{Vglut2}$→DMH projections, the body composition of both two groups of mice was assessed

with a minispec whole-body composition analyzer (Burker Minispec CMR LF50). And the fat and lean mass were normalized to total body mass. On the final day, the iBAT (interscapular brown adipose tissue) and iWAT (inguinal white adipose tissue) were collected, weighed, and frozen in liquid nitrogen from all mice after activation of the LPB$^{Vglut2}$ → DMH projections and then perfused transcardially to collect brain tissues.

## Cell-type specific retro-TRAP sequencing

The procedures were detailed in our previous reports[25]. Briefly, a recombinant Cre-dependent AAV plasmid expressing GFP-tagged ribosomal subunit L10a (AAV2-EF1a-DIO-EGFP-L10a) was gifted by Dr. Jeffrey Friedman and then packaged using rAAV2-retro capsids by Taitool. 300 nl of the viral aliquot was unilaterally injected into the DMH of Vglut2-IRES-Cre mice. Animals were anesthetized with isoflurane and sacrificed for immunoprecipitation four weeks after injection.

For TRAP experiments, brain slices (300 μm) containing the LPB regions were prepared from virus-injected mice and placed in ice-cold DEPC-PBS (VT 1200 S, Leica Microsystems, Germany). Then, the LPB regions were cut out with microsurgical forceps. Tissue from 6 brains was pooled for each experimental repeat, and three experimental repetitions were performed. Ribosomes were immunoprecipitated using anti-EGFP antibodies (Monoclonal Antibody Core Facility at Memorial Sloan-Kettering cancer center, an equal mixture of clones 19C8 and 19F7) that were conjugated to Protein L-coated magnetic beads (ThermoFisher Scientific) for 16-h at 4°C with rotating. An aliquot of input RNAs taken before immunoprecipitation and total immunoprecipitated RNAs were then purified and enriched with PCR to create the final cDNA library. Purified libraries were quantified and validated to confirm the insert size and calculate the mole concentration. The library construction and sequencing were performed by Shanghai Sinotech Genomics Corporation (China).

## TRAP-seq data analysis

The RNA quantification data after sequencing were analyzed as the following. To determine the statistical significance and fold enrichment, we divided the RNAs from the IP by those from the input for each gene (IP/Input). Moreover, the hits were narrowed down by only analyzing the PB-enriched gene list downloaded from Allen Institute (https://alleninstitute.org/; Tool one: MOUSE BRAIN CONNECTIVITY-source search-filter source structure: Parabrachial nucleus; Tool two: MOUSE BRAIN-Fine Structure Search: Parabrachial nucleus). The top candidate genes were then selected for further analysis.

## Reporting summary

Further information on research design is available in the Nature Portfolio Reporting Summary linked to this article.

## Data availability

Source data are provided with this paper and has been deposited in Figshare repository under accession code: https://doi.org/10.6084/m9.figshare.23652297.

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

## Acknowledgements

We thank Drs. Ji Hu, Zhenge Luo for sharing reagents; Dr. Xiaoming Li and the Molecular Imaging Core Facility (MICF) of School of Life Science and Technology, ShanghaiTech University for Microscopic imaging; Dr. Ying Xiong and the Molecular Cellular Core for slices and staining; all Shen lab members for valuable discussion. This study was funded by the National Key Research and Development Program of China (2019YFA0801900), National Nature Science Foundation of China (32122039 to W. Shen; 32100825 to W. Yang), Shanghai Science and Technology Committee of Shanghai City (21JC1404900 and 21XD1422700 to W. Shen), Shanghai Sailing Program (21YF1429800 to W. Yang), "Chen Guang" project supported by Shanghai Municipal Education Commission and Shanghai Education Development Foundation (20CG71 to W. Yang), Ministry of Science and Technology of China (2017YFA0205903), Shanghai Frontiers Science Center for Biomacromolecules and Precision Medicine at ShanghaiTech University, Open Research Fund of State Key Laboratory of Genetic Engineering-Fudan University (SKLGE-2303), and Singapore National Medical Research Council (MOH-000611-00 to Y. Fu) and Singapore National Research Foundation (NRF-NRFI07-2021-0001 to Y. Fu). We thank the staff members of the Animal Facility at ShanghaiTech University for providing technical support and assistance.

## Author contributions

W.Z.Y., H.X., X.D., and Q.Z. performed most of the experiments; Y.X., Z.L., and Z.Z. performed behavioral evaluations; S.Ch., R.H., J.C., S.Ca., J.S., H.S., Q.S., Q.X., and X.N. performed the immunostaining; X.J. performed retro-TRAP analysis; J.X. and W.Z. performed the electrophysiology; X.F. designed and performed a concurrent recording of BAT and core temperatures; W.Z.Y. and H.T. designed the graphical abstract; W.Z.Y., R.Z., X.X., H.W., Y.F., L.W., X.L., H.Y., Q.Y. T.Y., Q.S., and W.L.S. designed the experiments; W.Z.Y., H.X., X.D., Q.Z., and W.L.S. wrote the manuscript.

## Competing interests

The authors declare no competing interests.
