## [Peer Review File · Nature Communications]

A parabrachial-hypothalamic parallel circuit governs cold defense in miceREVIEWER COMMENTS

Reviewer #1 (Remarks to the Author):

In this study, Yang and colleagues used interdisciplinary techniques to illustrate the involvement of the lateral Parabrachial (LPB)- dorsomedial hypothalamus (DMH) in cold defense response for thermoregulation in mice, which complements previous findings that LPB to the preoptic area (POA) in regulating body temperature in mice. Specifically, aided by cre-dependent AAV tracing and c-Fos staining, the authors identified LPB Vglut-2 neurons projecting to POA and DMH are both responses to cold/warm exposure. Next, they used in vivo fiber photometry for neuronal calcium activities and observed LPB Vglut2-to-DMH pathways are robustly activated during gradients of cooling exposures. By employing a series of neuronal activity manipulations, they revealed that LPB→POA/DMH pathways form parallel roles in cold defense. Additionally, optogenetic activation shows LPB Vglut2→DMH pathway induces strong hyperthermia. Moreover, by selectively blocking the synaptic outputs of the downstream neurons using cell type-specific expression of TeNT, they found cold-defense responses depend on both Vglut2 and Vgat-expressing neurons in the DMH but not POA Vglut2-neurons. Notably, blocking both LPB→POA/DMH pathways exhibited a much stronger impairment in cold defense, suggesting a cumulative or additive effect of the two pathways. Activation of the LPB Vglut2→DMH pathway rapidly relieves hypothermia, increases iBAT thermogenesis, and suppresses body weight gain. Using retro-TRAP-seq, they identify DMH-projecting LPB SST neurons as cold-activated neurons. Finally, they showed LPB SST→DMH pathway increases Tcore via iBAT thermogenesis and is required for cold defense. Overall, this comprehensive study revealed an important central mechanism in understanding how neuronal circuits affect thermoregulation. The design of this study is straightforward, and the results are novel, exciting, and convincing. I have the following comments for the authors that will hopefully help them to improve the paper.

Major comments:

Their data shows that LPB neurons were activated in responding to cold exposure, and ~60% project to POA or DMH, and the authors claim that the two projecting pathways work in parallel in cold defense. I wondered if other parallel pathways work in concert with the POA and DMH projections in cold defense. This should either be tested or discussed.

It is convincing that the authors had shown multiple experiments that the activation LPB Vglut2-DMH pathway induces a strong cold defense response. In figure 4e-g the authors showed that DMH terminal activation could overcome LPB Vglut-2 soma hM4Di inhibition. Can the author show that chemogenetic inhibition can prevent the possible backpropagation of action potentials induced by the chemogenetic stimulation at the terminals? Moreover, I wonder if LPB Vglut-2 soma hM4Di inhibition would decrease Tcore? Although Figure 4g may argue against it, it is unclear why saline injection will cause an apparent slight decrease in the core temperature, similar to the CNO injection condition.

Several points need to be clarified for the optogenetic activation experiments. It is unclear what frequency and intensity are used in most paradigms. Have the authors evaluated the best photoactivation frequency in LPB-POA/DMH pathways? In Figure 5, what is the rationale for using 12 mw, 10 Hz 10 ms, 30-min per 2-h, 4 loops per day photoactivation? How robust could the LPB-DMH neuron follow this manipulation?

It is convincing that LPB SST-DMH plays an important function in cold defense. I would suggest that the author verify the colonization of SST and Vlut2 within in LPB region since SST are robust GABAergic neuron makers in most cases. If they do not totally overlap with Vglut2, are they GABAergic? Moreover, this may be beyond the scope of the current study. I wonder if SST, as a neuropeptide, is involved in the LPB SST-DMH pathway to modulate temperature regulation.

In figure 7, the author used SST-cre cross with LepR-cre or ChAT-cre mouse. Is it possible that DMH also has SST neurons?

Reviewer #2 (Remarks to the Author):

In this manuscript, the author characterized a new LPB→DMH pathway that is important for cold defensive behaviors. The battery of viral tools used in this manuscript is very impressive and the author provided solid data to support the LPB→DMH pathway in cold defense. However, results from this manuscript also raised some puzzling but important questions. These questions need to be address before we can put this new discovery in the context of our understanding of the central pathway for thermoregulation.

1. Why both LPB→DMH and LPB→MPO pathways are required for cold defensive behaviors? What is relationship between neurons in DMH and MPO that received inputs from LPB?
2. Similarly, why both DMHvglut2 vs. DMHVgat are required for cold induced defensive behavior?
3. In Fig.5F, why body weight no longer increase after light stimulation start in control mice? could it be the stress caused by long term light stimulation? why not using chemogenetic stimulation which could be less stressful?
4. Because LPBSST neurons also project to MPO? What is the percentage of LPBSST neurons→MPO neurons are Fos+ after cold stimulus?
5. There are so many markers for the DMH projecting LPBVglut2 neurons? are these neurons co-label same neurons or they label different population of neurons? this question is important because LPBSST neurons only represent less than 20% of Fos+ neurons after cold stimulus.
6. Are LPB is the only input to drive cold response in the DMH?

Minor point:

1. Because the limited retrograde efficiency of retrograde tracing virus, the conclusion of '20% projected to both regions' is likely underestimate.
2. Using terminal fiber photometry to study the kinetics of temperature response is not very meaningful, as they could be strongly modulate by varies autoreceptors located in the terminal.

Reviewer #3 (Remarks to the Author):

In this paper, the authors examine the hypothesis that cold responsive neurons in the lateral parabrachial nucleus (LPB) project to the dorsomedial nucleus of the hypothalamus, where they activate cold-defense pathways. I typically begin a review with a summary of all of the experiments that were done, however, that is nearly impossible for this paper. It is extremely long (over 7,000 words) and gives the results of over 40 experiments, many of which are very complicated. In most cases, there is no n given (in others where it is given, it is often very low numbers, such as 3 animals), and in no case is there given sufficient detail on injection placement (some of these injections have to miss their targets, but there is no information on how many total animals were done, how they picked the small number of animals they present, or what the anatomical controls, i.e., missed injections, showed), controls are not done for many key experiments, and it is rare to find any statistical analysis.

Having said that, I think this is an interesting story, which would be of interest to many scientists who work on thermoregulation. But it is impossible to evaluate the work critically in its current state because so much of the necessary information on the rigor of the experiments is missing. I would strongly encourage the authors, to include complete information for each experiment. This would include the power analysis that should have been done before the studies were started indicating the number of animals that should be in each group; the actual numbers of animals used in each experiment; how they chose which ones to present in the paper; details about the actual results in the animals that are included and the ones that are not included, as well as controls; how they did the statistics; and what the statistical findings were. This is particularly important for experiments involving stereotaxic injections, some of which will miss their intended target. How were those cases identified? They should be analyzed by someone who does not know the physiological results. The ones that hit the target should be analyzed separately from those that missed the target, which then serve as anatomical controls. But you need to present heat maps showing the actual injection placements in both sets of animals.

Responses to Reviewers

Reviewer #1 (Remarks to the Author):

In this study, Yang and colleagues used interdisciplinary techniques to illustrate the involvement of the lateral Parabrachial (LPB)- dorsomedial hypothalamus (DMH) in cold defense response for thermoregulation in mice, which complements previous findings that LPB to the preoptic area (POA) in regulating body temperature in mice. Specifically, aided by cre-dependent AAV tracing and c-Fos staining, the authors identified LPB Vglut-2 neurons projecting to POA and DMH are both responses to cold/warm exposure. Next, they used in vivo fiber photometry for neuronal calcium activities and observed LPB Vglut2-to-DMH pathways are robustly activated during gradients of cooling exposures. By employing a series of neuronal activity manipulations, they revealed that LPB→POA/DMH pathways form parallel roles in cold defense. Additionally, optogenetic activation shows LPB^{Vglut2}→DMH pathway induces strong hyperthermia. Moreover, by selectively blocking the synaptic outputs of the downstream neurons using cell type-specific expression of TeNT, they found cold-defense responses depend on both Vglut2 and Vgat-expressing neurons in the DMH but not POA^{Vglut2}-neurons. Notably, blocking both LPB→POA/DMH pathways exhibited a much stronger impairment in cold defense, suggesting a cumulative or additive effect of the two pathways. Activation of the LPB^{Vglut2}→DMH pathway rapidly relieves hypothermia, increases iBAT thermogenesis, and suppresses body weight gain. Using retro-TRAP-seq, they identify DMH-projecting LPB SST neurons as cold-activated neurons. Finally, they showed LPB SST→DMH pathway increases T_{core} via iBAT thermogenesis and is required for cold defense. Overall, this comprehensive study revealed an important central mechanism in understanding how neuronal circuits affect thermoregulation. The design of this study is straightforward, and the results are novel, exciting, and convincing. I have the following comments for the authors that will hopefully help them to improve the paper.

R: Thanks for the appreciation of our efforts in the dissection of the cold-defense circuit.

Major comments:

Their data shows that LPB neurons were activated in responding to cold
exposure, and ~60% project to POA or DMH, and the authors claim that the
two projecting pathways work in parallel in cold defense. I wondered if other
parallel pathways work in concert with the POA and DMH projections in cold
defense. This should either be tested or discussed.

R: We appreciate the attention given to this matter. The DMH-projecting
LPB^{Vglut2} neurons were found to project to several brain regions, including the
LH and VMH, as shown in **Extended Data Fig. 3**. Nevertheless, the LPB→LH
projection was demonstrated to be dispensable for cold-induced thermogenesis,
as evidenced by **Cited Fig. 1L (1)**. Furthermore, although LPB neurons
projecting to the VMH were predominantly located in the LPBc region, as
depicted in **Cited Fig. 2b, c. (2)**, they were not co-localized with the LPB_{el}
neurons that were activated by cold stimuli. Thus, we have included a
discussion in **lines 606-613**, noting that "it is noteworthy that the LPB neurons
projecting to the POA/DMH region also have projections to other brain regions
involved in thermoregulation, such as the LH and VMH (**Extended Data Fig.**
**3**). However, the LPB→LH projection was not found to be necessary for cold-
induced thermogenesis (1), and the VMH-projecting LPB neurons were not
responsive to cold stimuli (2, 3). Further investigation is necessary to ascertain
the existence of parallel pathways that operate in conjunction with the
LPB→POA/DMH projections."

**Extended Data Fig. 3 | Projection pattern of DMH-projecting LPB^{Vglut2}**
**neurons throughout the brain.** (a) Scheme for mapping the axonal
projections of DMH-projecting LPB^{Vglut2} neurons. For labeling DMH-projecting
LPB^{Vglut2} neurons, retrograde AAVs carrying Cre-dependent FlpO were injected
in the DMH, which drove the expression of FlpO-dependent ChR2-eYFP in the

LPB. A red tracer (CTB647) was co-injected into the DMH to indicate the
 injection sites. (b) Representative image showing the injection sites in the DMH
 viewed by red tracer (CTB647). (c) Representative images showing ChR2-
 eYFP expression in DMH-projecting LPB^{Vglut2} neurons in the LPB. (d)
 Representative images are showing ChR2-eYFP expression in axonal
 terminals at various brain sites.

**Cited Fig. 1** | (L) LPB-LH projection inhibition had no effect on iBAT
 temperature during cold challenge. Cited from (1).

**Cited Fig. 2** | (b) LPB lepR positive neurons mainly distributed in LPBc and
 send projections to VMH (c). Cited from (2).

It is convincing that the authors had shown multiple experiments that the
 activation LPB^{Vglut2}-DMH pathway induces a strong cold defense response. In
 figure 4e-g the authors showed that DMH terminal activation could overcome
 LPB^{Vglut-2} soma hM4Di inhibition. Can the author show that chemogenetic
 inhibition can prevent the possible backpropagation of action potentials induced
 by the chemogenetic (optogenetic?) stimulation at the terminals?

**R:** We acknowledge the potential concern of backpropagation of action
 potentials in optogenetic experiments. However, the limited literature on this
 topic suggests that this issue may be overestimated. To address this concern,
 it would be ideal for us to conduct electrophysiological recordings of
 backpropagation of action potentials in LPB brain slices. As simultaneous
 activation of DMH terminals and recording of backpropagation of action

potentials in LPB brain slices is technically challenging due to the long distance
between LPB and DMH, we directly recorded calcium activity in vivo. We used
red-shift opto-tools ChrimsonR to estimate whether photoactivated DMH
terminals were strong enough to evoke calcium activity in LPB somata. Co-
expression of GCaMP6s, DREADD-Gi, and ChrimsonR was done in the LPB,
followed by the activation of DMH terminals using a 589-nm laser (**Reviewer**
**only Fig. 1a**). Ca²⁺ activity was recorded in the LPB, and we confirmed that
DMH terminal photostimulation could induce hyperactivity, and DREADD-Gi
inhibition could substantially reduce LPB somatic neural calcium activity
(**Reviewer only Fig. 1b-c**). However, we did not observe any calcium signal
changes in LPB somata due to DMH terminal photoactivation (in short or long
terms) (**Extended Data Fig. 7c**). Thus, we conclude that backpropagated
action potentials are too weak to be detected by in vivo calcium signals.
Additionally, we argue that if any undetected backpropagation of action
potentials exists, it should be blocked by DREADD-Gi inhibition since Gi
inhibition has a larger impact on LPB neural activity than that of DMH terminal
photoactivation.

**Reviewer only Fig. 1** | (a) Simultaneously expressing the GCaMP6s,
DREADD-Gi and ChrimsonR in the LPB, then the DMH terminals were
simultaneously activated using a 589nm laser and Ca²⁺ activity was recorded
in the LPB; (b) Photoactivation of DMH terminal with ChrimsonR result in
hyperactivity. (c) LPB neurons' calcium activity was inhibited by hM4Di after
CNO injection.

C LPB Soma Recording while DMH Terminal Activation

**Extended Data Fig. 7** | (c) Scheme for simultaneously expressing the
 GCaMP6s and ChrimsonR in the LPB, then the DMH terminals were
 simultaneously activated using a 589nm laser and Ca^{2+} activity was recorded
 in the LPB (left panel). No obvious calcium signal changes were recorded in
 LPB after activating DMH's terminals (n = 5 mice). The blue pulse line indicates
 light stimulation (right panel). Light pattern: 589 nm, 6 mW, 10 Hz, 10 ms, 2-s
 on followed by 2-s off, with the cycles repeating for 10 min.

Moreover, I wonder if LPB V_{glut-2} soma hM4Di inhibition would decrease T_{core} ?

**R:** In brief, we did not find that acute inhibition of LPB V_{glut2} soma with hM4Di
 changed T_{core} . As previously reported (**Cited Fig. 3**) (3), broad activation of
 LPB V_{glut2} soma can lead to hypothermia or hyperthermia, depending on the
 stimulation frequency. Therefore, it is not surprising that broad inhibition of
 LPB V_{glut2} neurons did not alter T_{core} , as there are mixed hypothermic and
 hyperthermic effects in this region. Our results show that the acute increase of
 T_{core} after Sal/CNO injection was induced by stress, as indicated in the baseline
 data before injection (**Fig. 4g**).

**Cited Fig. 3** | The change of T_{core} after photoactivation of LPB V_{glut2} soma (D)
 and VMPO terminal (E) with different frequencies. Light pattern: 473 nm, 6 mW,
 5/10/20/40 Hz, 10 ms, 2-s on followed by 2-s off, with the cycles repeating for
 30 min.

**Fig. 4 | (g)** Changes in T_{core} after activating the $\text{LPB}^{\text{Vglut2}} \rightarrow \text{DMH}$ projection while
 blocking LPB neurons (n = 11 mice). CNO was injected at -60 min to silence
 neurons as indicated (i.p., 10 mg/kg) and saline was used as the control.

Although Figure 4g may argue against it, it is unclear why saline injection will
 cause an apparent slight decrease in the core temperature, similar to the CNO
 injection condition.

**R:** As previously mentioned (**Fig. 4g**), the observed phenomenon involves the
 restoration of normothermia following an acute increase in T_{core} induced by
 Sal/CNO injection-induced stress and is not a hypothermia response.

Several points need to be clarified for the optogenetic activation experiments.
 It is unclear what frequency and intensity are used in most paradigms. Have
 the authors evaluated the best photoactivation frequency in LPB-POA/DMH
 pathways?

**R:** We have provided information regarding the frequency and intensity of
 optogenetic activation experiments in the figure legend (6 or 12mW 10 Hz, 10
 153 ms, 2-s on followed by 2-s off, with the cycles repeating for 30 min). We did
 evaluate the photoactivation frequency for both pathways and published the
 data for stimulation of the LPB-POA pathway (3). We did observe that the
 hypothermia induced by terminal activation of the LPB \rightarrow POA pathway
 increased as the photoactivation frequency increased (**Cited Fig. 3**) (3). In the
 case of the LPB \rightarrow DMH pathway, we found that hyperthermia induced by 10-
 159 Hz photoactivation was greater than that induced by 5 Hz and 20 Hz, although
 this difference was not statistically significant. We have now added this data to
 **Extended Data Fig. 7a, b.**

**Extended Data Fig. 7** | (a-b) The change of T_{core} after photoactivation of
 LPB^{Vglut2} DMH terminal with different frequencies. Light pattern: 473 nm, 12 mW,
 5/10/20 Hz, 10 ms, 2-s on followed by 2-s off, with the cycles repeating for 30
 166 min.

In Figure 5, what is the rationale for using 12 mw, 10 Hz 10 ms, 30-min per 2-
 169 h, 4 loops per day photoactivation? How robust could the LPB-DMH neuron
 follow this manipulation?

**R:** In our study, experiments were primarily conducted during the active phase
 of the mice (9:00 am- 9:00 pm). Each mouse required fiber attachment and
 equipment setup, which typically took 1-2 hours. Therefore, the available time
 for photoactivation was limited to 9-10 hours. In addition, to ensure adequate
 recovery time from the previous photoactivation (which lasted 1-1.5 hours),
 each photoactivation session was limited to 30 minutes. Thus, we could only
 perform a maximum of 4 photoactivation sessions per day. As illustrated in **Fig.**
 **5e**, the LPB-DMH neurons exhibited a robust response to optogenetic
 manipulation, resulting in a 1-1.5°C increase in T_{core} . We have included this data
 in our figures for reference.

**Fig. 5** | (e) The change of T_{core} after 4 times photoactivation (left panel) of
 LPB^{Vglut2} DMH terminal and the comparison (right panel) of T_{core} change for
 each photoactivation.

It is convincing that LPB^{SST}-DMH plays an important function in cold defense. I
 would suggest that the author verify the colonization of SST and Vglut2 within
 in LPB region since SST are robust GABAergic neuron makers in most cases.
 If they do not totally overlap with Vglut2, are they GABAergic? Moreover, this
 may be beyond the scope of the current study. I wonder if SST, as a
 neuropeptide, is involved in the LPB SST-DMH pathway to modulate
 temperature regulation.

R: In response to the first question, we're glad that we have the same interest
 as the reviewer. The relevant data had already been included in original
 **Extended Data Fig. 8b**. Our analysis has confirmed that the LPB^{SST} neurons
 are primarily glutamatergic, as confirmed by glutamate staining. This finding is
 consistent with previous studies(3-6) reporting that the vast majority of neurons
 (>98%) in LPB are also glutamatergic, as evidenced by the sequencing of the
 SST gene from DMH-projecting LPB^{Vglut2} neurons (**Fig. 6a-c**). As SST is
 primarily used as a GABAergic neuron marker in the cortex, it is worth noting
 that in other regions such as the hypothalamus, SST neurons are capable of
 exhibiting both glutamatergic and GABAergic phenotypes (7). Therefore, we did
 not present this data in the main figure.

**Extended Data Fig. 8 | (b)** Overlap between SST-IRES-Cre labeled neurons
 (Tdt⁺) in the LPB and the following immuno-positive neurons and glutamate.
 SST-IRES-Cre mice were crossed with Ai14 (Rosa-CAG-LSL-tdTomato-WPRE)
 to label SST-IRES-Cre in the LPB.

**Fig. 6** | (a) Projection-specific transcriptomic analysis (retro-TRAP), where
GFP-tagged translational ribosomes from DMH-projecting LPB^{Vglut2} neurons
were immunoprecipitated and associated mRNAs were sequenced. Retrograde
tracing virus carrying Cre-dependent GFP-tagged ribosomal protein L10 (AAV-
Retro-hEF1a-FLEX-GFPL10) was injected into the DMH, which traveled to the
LPB and expressed GFPL10 after recombination by Vglut2-IRES-Cre. SST,
somatostatin. (b) Volcano plots (q value versus log2 fold change) for LPB
mRNAs after retro-TRAP sequencing. (c) Retro-TRAP fold enrichment (IP/Input)
for PB-expressed genes downloaded from the Allen Institute.

In regard to the second question, it is worth noting that the SST peptide was
administered into the brain over 50 years ago, and numerous studies (8-11)
have since reported that this injection can induce hyperthermia, particularly
when targeting the DMH. It would be interesting to explore the potential role of
endogenous SST peptide in thermoregulation in future research.

In figure 7, the author used SST-cre cross with LepR-cre or ChAT-cre mouse.
Is it possible that DMH also has SST neurons?

R: We would like to acknowledge that the data referred to by the reviewer has
indeed been included in **Fig. 7q** of our manuscript. In addition, we have
provided clarification in **lines 519-521** of our manuscript by stating that "since
SST-Cre is also expressed in the DMH (**Fig. 7q, left panel**).". Furthermore, our
statement "blocking DMH^{ChAT+SST} neurons had no effect" serves to exclude the
role of DMH^{SST} neurons in this manipulation.

**Fig. 7** | (q) Representative SST-Cre & Tdt expression in the DMH.

**Reviewer #2 (Remarks to the Author):**

In this manuscript, the author characterized a new LPB→DMH pathway that is
important for cold defensive behaviors. The battery of viral tools used in this
manuscript is very impressive and the author provided solid data to support the
LPB→DMH pathway in cold defense. However, results from this manuscript
also raised some puzzling but important questions. These questions need to be
addressed before we can put this new discovery in the context of our
understanding of the central pathway for thermoregulation.

1. Why both LPB→DMH and LPB→MPO pathways are required for cold
defensive behaviors? What is the relationship between neurons in DMH and
MPO that received inputs from LPB?

**R:** For the first question why two parallel pathways are required in cold defenses.
As we wrote in the discussion section: “The evolution of parallel neural circuits
in cold defense not only enables resilience to hypothermia but also provides a
scalable, robust, and efficient network in heat production when both pathways
are recruited.” (lines 618-621). In line with this hypothesis, we have presented
data to show that activation of the LPB→DMH pathway is powerful enough to
reverse cold-induced hypothermia (Fig. 4p-r). Furthermore, we have shown
that both pathways are additively or synergistically required to boost cold
defense (Fig. 3g-k).

**Fig. 4** | (p-q) Changes in T_{core} after photoactivation of the LPB^{Vglut2}→DMH
projection under different T_a (6, 24, 30°C; $n = 6$ mice each) (p). The maximum
ΔT_{core} during photoactivation was quantified in (q). (r) Normalized heat
production of mice after photoactivation of the LPB^{Vglut2}→DMH projection under
different T_a as indicated. Heat production was normalized by the formula (T_{core}
$-T_a$) / ΔT_{core} as reported.

**Fig. 3 |** (g) Blocking LPB-innervating POA/DMH neurons or both using TeNT.
 The detailed scheme is shown in Extended Data Fig. 5a. Briefly, anterograde
 transsynaptic Cre carried by AAV1 (AAV1-hSyn-Cre) was injected in the LPB to
 drive expression of Cre-dependent TeNT injected in either the POA (POA^{LPB}
 blocking), or the DMH (DMH^{LPB} blocking), or both (co-blocking). Cre-dependent
 GFP co-injected in both the POA and DMH was used as the control (GFP
 control). (h-k) T_{core} changes in response to a series of cold exposures, namely
 29→22°C (h), 29→16°C (i), 29→10°C (j), and 29→4°C (k), after blocking
 POA^{LPB}, DMH^{LPB}, or both types of neurons (n = 10 mice for GFP and DMH^{LPB}
 blocking group; n = 9 mice for POA^{LPB} blocking group; n = 6 mice for the co-
 blocking group). *, DMH^{LPB} blocking vs. GFP; \$, DMH^{LPB} vs. POA^{LPB} blocking;
 #, POA^{LPB} vs. co-blocking.

For the second question regarding the relationship between neurons in DMH
 and MnPO/VMPO that received inputs from LPB. As summarized by Kazuhiro
 (12) (Cited Fig. 4), LPB-innervating MnPO/VMPO neurons contain separate
 subgroups of neurons for either cold or warm defenses and therefore are
 responsible for both cold and warm defenses. In contrast, LPB-innervating
 DMH neurons are only responsible for cold defense. We also verified this
 conclusion by blocking MnPO/VMPO-projecting or DMH-projecting LPB^{Vglut2}
 neurons (Reviewer only Fig. 2a-b). Both blockings impaired cold defenses but
 only blocking MnPO/VMPO-projecting LPB^{Vglut2} neurons impaired warm
 defense (Reviewer only Fig. 2c-d). Taken together, LPB-innervating
 MnPO/VMPO and DMH neurons function in parallel in cold defense. In contrast,
 according to previous numerous literature (12-14), DMH neurons, including

LPB-innervating DMH neurons, are expected to act downstream of LPB-
 innervating MnPO/VMPO neurons to form a feed-forward pathway in warm
 defense.

 **Cited Fig. 4 | (b) A model of the POA local circuit that controls effector**
 **responses to thermal and infection stresses.** Cutaneous warm-sensory
 inputs from the dorsal part of the lateral parabrachial nucleus (LPBd) activate
 glutamatergic interneurons in the MnPO (blue-shaded area), which then
 activate GABAergic projection neurons in the MPA, the MnPO and the ventral
 part of the lateral preoptic area (vLPO) (red shaded areas). These projection
 neurons inhibit excitatory neurons in the dorsomedial hypothalamus (DMH) and
 rostral medullary raphe region (rMR) that otherwise drive cold-defensive
 responses. (Cited from (12))

 **Reviewer only Fig. 2 | (a) Scheme for blocking DMH-projecting LPB^{Vglut2}**
 **neurons.** For blocking DMH-projecting LPB^{Vglut2} neurons, retrograde AAVs
 carrying Cre-dependent FlpO were injected in the DMH, which drove the
 expression of FlpO-dependent TeNT in the LPB. (b) Scheme for blocking POA-
 projecting LPB^{Vglut2} neurons. For blocking POA-projecting LPB^{Vglut2} neurons,
 retrograde AAVs carrying Cre-dependent FlpO were injected in the POA, which
 drove the expression of FlpO-dependent TeNT in the LPB. (c-d) Changes in
 T_{core} after blocking of DMH-projecting LPB^{Vglut2} or POA-projecting LPB^{Vglut2}

neurons during the cold (c) or warm (d) challenge. #, mCherry vs DMH-
projecting TeNT; \$, mCherry vs POA-projecting TeNT; *, DMH-projecting TeNT
vs POA-projecting TeNT.

2. Similarly, why both DMH^{Vglut2} vs. DMH^{Vgat} are required for cold induced
defensive behavior?

R: In our previous publication in *PNAS* (15), we clearly demonstrated that both
DMH^{Vglut2} vs. DMH^{Vgat} are important for cold defense since they both could
bidirectionally control T_{core} upon activation/inhibition. By comparing the whole-
brain projection data of $DMH^{Vglut2/Vgat}$ neurons from the Allen Brain Institute, we
found that DMH^{Vgat} neurons mainly project to the POA with few projections to
RPa (rMR), while DMH^{Vglut2} neurons send significantly more projections to RPa
than DMH^{Vgat} neurons (Cited Fig. 5). Therefore, we proposed that DMH^{Vglut2}
neurons primarily provide excitatory input to premotor neurons in the RPa to
stimulate thermogenesis, while DMH^{Vgat} neurons may mainly suppress POA
hypothalamic neurons to inhibit heat loss. Working together, these two neuron
types could maximize heat production to defend against cold. In line with the
evidence, our preliminary data suggest that the projection from the DMH^{Vgat} to
the vLPO indeed could increase T_{core} (Reviewer only Fig. 3)

Cited Fig. 5 | The terminal distribution of DMH^{Vgat} (a) and DMH^{Vglut2} neurons (b)
in the POA and RPa (downloaded from Allen brain institute).

**Reviewer only Fig. 3** | (a) Scheme for photoactivation of neural terminals of
 DMH^{Vgat} neurons in the vLPO using ChR2. (b) Representative expression of
 ChR2 terminals from DMH^{Vgat} neurons in the vLPO. (c) ChR2-expressing
 terminals were activated with a 2-ms blue light to elicit inhibitory postsynaptic
 currents (IPSCs) in vLPO neurons. IPSCs were blocked by bicuculline (bic.)
 and recovered partially after washing. Shadowed areas were SD. (d-e)
 Changes of T_{core} (d), and physical activity (e) after neural terminals of DMH^{Vgat}
 neurons in the vLPO (ChR2, n = 6 mice; GFP, n = 4). Light pattern: 473 nm, 6
 345 mW, 20 Hz, 10 ms, 2-s on 2-s off, 60 min.

3. In Fig.5F, why body weight no longer increases after light stimulation start in
 control mice? could it be the stress caused by long term light stimulation? why
 not using chemogenetic stimulation which could be less stressful?

**R:** We noticed that the stress caused by fiber attachment during long-term light
 stimulation might curb the body weight gains. Therefore, we wrote in **line 412-**
 **415:** "Although the photostimulation procedure itself appeared to curb weight
 gains in control mice, photoactivation of the LPB^{Vglut2}→DMH projection further
 reduced body weight without affecting cumulative food intake (**Fig. 5f-h**).". As
 suggested by this reviewer, the chemogenetics could be suitable for long-term
 stimulation in soma stimulations. However, using chemogenetic to activate the
 terminal requires repetitive injection of CNO to the DMH through a cannula,
 which might also cause stress. Therefore, we did not adopt the chemogenetic
 approaches. We hope this reviewer would agree with us and understand the
 limitations of both approaches.

**Fig. 5** | (f) Mice after two weeks of photoactivation (left, ChR2 group; right, GFP
 group). (g-h) Changes in body weight (g) and cumulative food intake (h) during
 two weeks of photoactivation (ChR2, n = 9 mice; GFP, n = 5 mice, a total of 10
 mice for ChR2 and 6 mice for GFP were injected, 1 mouse for ChR2 and 1
 mouse for GFP were excluded from the final analysis due to death). The light
 pattern is shown in (d).

4. Because LPB^{SST} neurons also project to MPO? What is the percentage of
 LPB^{SST} neurons → MPO neurons are Fos⁺ after cold stimulus?

R: Thanks for this careful notice. As shown below, about 20 percent of POA-
 projecting LPB^{SST} neurons are Fos⁺ after cold stimulus (**Extended Data Fig.**
 **9b**). In contrast, more (~40%, **Extended Data Fig. 9a**) POA-projecting LPB^{SST}
 neurons are activated after a heat stimulus. Nevertheless, we photoactivated
 the LPB^{SST} → POA projections and found a hypothermia phenotype (**Extended**
 **Data Fig. 9d**). These data resembled the photoactivation phenotypes seen
 after bulk activation of LPB^{Vglut2} → POA projections, which causes hypothermia
 only. We added this data to **Extended Data Fig. 9a-d**.

**Extended Data Fig. 9** | (a-b) Overlap between POA-projecting LPB^{SST} neurons
 and heat-induced cFos (a) or cold-induced cFos (b) (n = 3 mice each). To label
 POA-projecting LPB^{SST} neurons, we injected retrograde AAVs carrying Cre-
 dependent GFPL10 (AAV-Retro-CAG-Flex-GFPL10) in the VMPO of SST-
 IRES-Cre mice, which drove the expression of GFPL10 in the LPB. Merged

cells were indicated by white arrows. (c) Design to activate the LPB^{SST}→POA
projection via photostimulating of LPB^{SST} & ChR2 terminals in the VMPO. The
representative expression of ChR2-eYFP in the POA is shown in right. (d)
Changes in T_{core} after photoactivation of the LPB^{SST}→POA projection. (n = 6
mice). Light pattern: 473 nm, 6 mW, 10 Hz, 10 ms, 2-s on followed by 2-s off,
with the cycles repeating for 30 min.

5. There are so many markers for the DMH-projecting LPB^{Vglut2} neurons? are
these neurons co-label same neurons or they label different population of
neurons? this question is important because LPB^{SST} neurons only represent
less than 20% of Fos⁺ neurons after cold stimulus.

R: We thank this reviewer for noticing many “markers” for the DMH-projecting
LPB^{Vglut2} neurons from our Retro-TRAP sequencing data. We actually studied
several of them, including n4bp2os, Gal, SST, TH, CCK, and Pdyn. Since a Cre
strain was not available to label n4bp2os+ neurons, we did not further
investigate this marker. We reported previously that there were few Gal-Cre⁺
and TH⁺ neurons in the LPB (**Cited Fig. 6C, D**) (3). Instead, we found a cluster
of Gal-Cre⁺ neurons in the laterodorsal tegmental nucleus and the locus
coeruleus (**Cited Fig. 6C**), and a cluster of TH⁺ neurons in the locus coeruleus
(**Cited Fig. 6D**). Their enrichment might presumably be due to tissue
contamination from nearby areas. For CCK and Pdyn, we reported before that
there were no changes in T_{core} after activation of LPB^{Pdyn/CCK} terminals in the
DMH (**Cited Fig. 6 E, F and H, I**) (3).

Nevertheless, we agree with the reviewer that there should exist other
important markers for cold defense in the LPB-DMH pathway. Therefore, we
wrote in **line 633-638**: “We showed that the LPB^{SST}→DMH^{LepR} pathway
governs iBAT thermogenesis, suggesting a genetically defined projection
controls specific cold defense activities. We reasonably speculate that other
cold defense activities, including heart rate and muscle shivering, are also
controlled by genetically defined neural projections.”. Significant effort was
needed to identify other genetic markers for cold defense in the LPB-DMH
pathway.

**Cited Fig. 6** | (C) GFP expression of Galanin-Cre & LSL-GFP10 mice in the
 LPB, LDTg and LC. (D) The staining of warm-induced cFos and TH (tyrosine
 hydroxylase) in the LPB showed nearly no TH⁺ soma in the LPB. (E) Expression
 of ChIEF from LPB^{CCK} neural terminals in the DMH. (F) Changes of T_{core} after
 photoactivation of LPB^{CCK} neural terminals in the DMH. (H) Expression of
 ChIEF from LPB^{Pdyn} neural terminals in the DMH. (I) Changes of T_{core} after
 photoactivation of LPB^{Pdyn} neural terminals in the DMH.

6. Are LPB is the only input to drive cold response in the DMH?

R: There are other inputs to the DMH besides the LPB. The POA has long been
 considered the primary upstream region for driving cold responses (12, 13, 16).
 Recent studies have identified POA^{BRS3} neurons that can drive cold response
 through the DMH (**Cited Fig. 7B**) (17). Additionally, the DP/DTT provides input
 to the DMH to drive thermogenic response during psychological stress (**Cited**
 **Fig. 8F**) (18). However, further studies are required to investigate whether the
 DP/DTT can also be activated by cold stress. Whether these different inputs
 act on the same or different DMH neural subtypes is unclear. Yet, based on the
 tracing studies, DMH^{BRS3} received much more input from the POA than from the
 LPB (**Cited Fig. 9**, (14)). Thus, it is reasonable to speculate that different inputs
 may act on slightly different DMH neural subtypes to promote thermogenesis.

**Cited Fig. 7** | (B) Optogenetic stimulation of POA^{brs3}→DMH axons increases
 T_{core}.

**Cited Fig. 8** | (F) The DP/DTT integrates signals from multiple forebrain regions
 processing stress and emotion and then provides a glutamatergic (Glu) master
 signal to the DMH to excite neuronal groups controlling different effectors.

**Cited Fig. 9 | DMH^{Brs3}→RPa neurons receive input from POA and other**
 **nuclei.** (a) Schematic of projection-specific rabies tracing. b, Brs3-Cre; Ai14
 mice injected with AAV-DIO-TVA-mCherry (TVA, avian tumor virus receptor A)
 in the DMH and EnvA-G-deleted-Rabies-GFP in the RPa, showing the dDMH–
 DHA localization of DMH^{Brs3}→ RPa neurons. (c–i) Brs3-Cre mice were injected
 with Flex-TVA-mCherry and Flex-RG (RG, rabies glycoprotein) viruses in the
 DMH and EnvA-G-deleted-Rabies-GFP in the RPa. (c,d) DMH showing
 DMH^{Brs3}→ RPa starter neurons expressing TVA-mCherry and GFP. (d) Higher
 magnification of inset. (e–h) Examples of regions with higher numbers of input
 neurons, which express only GFP. (i) Areas with input neurons to DMH^{Brs3}→
 RPa neurons, percentage of total.

Minor point:

1. Because the limited retrograde efficiency of retrograde tracing virus, the
conclusion of '20% projected to both regions' is likely underestimate.

R: We express our gratitude to the reviewer for highlighting the limitations of
our study with regard to retrograde efficiency, which we acknowledge.
Accordingly, we have added a statement in **lines 1151-1152**: "It is noteworthy
that this percentage might be underestimated due to the limited retrograde
efficiency."

2. Using terminal fiber photometry to study the kinetics of temperature response
is not very meaningful, as they could be strongly modulated by varies
autoreceptors located in the terminal.

R: We appreciate the reviewer's valuable information. We would also like to
note that we used retrograde labeling with GCaMP6s to record projection-
specific soma calcium responses (**Fig. 2p-t**), which provides complementary
data to the terminal recording data.

**Fig. 2** | (p-q) Recording from DMH-projecting LPB^{Vglut2} neurons (p) and
representative expression of GCaMP6s (left) and summary of fiber tracts
(shown as blue dots, right) (q). Retrograde traveling AAV-Retro-hSyn-Flex-
GCaMP6s were injected in the DMH of Vglut2-Cre mice, which traveled to the
LPB to drive GCaMP6s expression in the soma of LPB^{Vglut2} neurons. (r) Calcium
dynamics of DMH-projecting LPB^{Vglut2} neurons in response to floor warming (25
→38°C) or cooling (25→10°C) (n = 6 mice). (s) Peak $\Delta F/F_0$ values during floor
warming or cooling (n = 6 mice, average $\Delta F/F_0$ of 5 trials per mouse). (t) Mean
$\Delta F/F_0$ values during the steady phase after floor warming or cooling (n = 6 mice,
average $\Delta F/F_0$ of 5 trials per mouse).

**Reviewer #3 (Remarks to the Author):**

In this paper, the authors examine the hypothesis that cold responsive neurons
in the lateral parabrachial nucleus (LPB) project to the dorsomedial nucleus of
the hypothalamus, where they activate cold-defense pathways. I typically begin
a review with a summary of all of the experiments that were done, however,
that is nearly impossible for this paper. It is extremely long (over 7,000 words)
and gives the results of over 40 experiments, many of which are very
complicated. In most cases, there is no n given (in others where it is given, it is
often very low numbers, such as 3 animals), and in no case is there given
sufficient detail on injection placement (some of these injections have to miss
their targets, but there is no information on how many total animals were done,
how they picked the small number of animals they present, or what the
anatomical controls, i.e., missed injections, showed), controls are not done for
many key experiments, and it is rare to find any statistical analysis.

R: We appreciate the reviewer for bringing up these points here and I believed
these points have been brought up by this reviewer when we submitted to
another journal. We actually have made significant efforts to reduce the
complexity of the manuscript before the submission here. Now, we have made
additional efforts to improve the clarity of the manuscript and included the
technical and statistical details mentioned above.

We have listed numbers in the figure or plotted data in dots to show
numbers and had previously summarized **all numbers and detailed statistics**
**in Extended Data Table 2**. Now, we clearly marked the numbers on the figures
and described the statistics clearly in the figure legends as well. We used a
minimum of n = 6 mice for behavioral testing, n = 3 mice for staining analysis,
and n = 4 for EMG recording. After behavioral tests were finished, mice were
perfused to check the virus expression and fiber insertion. Data from mice that
showed little or no viral expression or had a fiber insertion that missed the target
(often 0-20%) were excluded from the analysis (please see **Extended Data Fig.**
**6, Reviewer only Fig. 4** and methods **lines 69-71** for more information).

a

b

One mouse that missed target of photoactivation

**Extended Data Fig. 6** | (a) Representative images demonstrating the
expression of ChR2 in LPB^{Vglut2} neurons, fiber insert positions and resulting
T_{core} changes after photoactivation LPB^{Vglut2}→DMH that hit the target. (b) Data
from one mouse exhibiting missed fiber insert position and T_{core} changes
following photoactivation. AHN, anterior hypothalamic nucleus.

**Reviewer only Fig. 4** | (a) Summary of fiber tracts that hit the target and miss
 target (shown as blue dots) of LPB^{SST}→DMH. (b) Data from two mice exhibiting
 miss fiber insert positions and T_{core} changes following photoactivation. PVH,
 Paraventricular hypothalamic nucleus.

For viral expression, besides showing representative virus expression
 images and optical fiber locations, we included the anatomical map of viral
 expression and optical fiber location maps (Fig. 1d, 2b, 2q, 4b, 7b, and so on).
 For example, we provided the anatomy map of retrograde tracers injected in
 the POA and the DMH (Extended Data Fig. 2a). We also provided the
 expression map of these retrograde tracers in the LPB (Extended Data Fig.
 2b) and Heatmaps of TeNT expression and fiber tracts (shown as green dots)
 at different Bregma sites from key experimental mice (Extended Data Fig. 5b
 and Extended Data Fig. 7n).

As for the concern of control experiments, we indeed provided proper
 controls for all the experiments. The design of this study has been recognized
 by reviewers #1 and #2. “The design of this study is straightforward, and the
 results are novel, exciting, and convincing” by reviewer #1 and “The battery of
 viral tools used in this manuscript is very impressive and the author provided
 solid data to ...” by reviewer #2.

Together, we thank this reviewer for witnessing the improvement of this
 manuscript and hope this reviewer would appreciate our efforts made to
 address the issues.

**Extended Data Fig. 2 |** Mapping the collateral projections of LPB^{Vglut2}
 neurons to the POA (MnPO & VMPO) and the DMH.

**Extended Data Fig. 5 |** (b) Heatmaps of TeNT expression after blocking LPB-
 innervating POA or DMH neurons.

n

Heatmaps of TeNT expression and fiber tracts of all blocking mice above

**Extended Data Fig. 7 |** (n) Heatmaps of TeNT expression and fiber tract ends
(shown as green dots) at different Bregma sites from all experimental mice.
DMH^{Vglut2} blocking, n = 9 mice; DMH^{Vgat} blocking, n = 7 mice; POA blocking, n
= 7 mice. The relative scale for the expression intensity (measured by
fluorescence intensity) was shown on the right.

Having said that, I think this is an interesting story, which would be of interest
to many scientists who work on thermoregulation. But it is impossible to
evaluate the work critically in its current state because so much of the
necessary information on the rigor of the experiments is missing. I would
strongly encourage the authors, to include complete information for each
experiment.

**R:** We express our gratitude for the reviewer's interest in our work. We have
taken the reviewer's feedback into careful consideration and have made
significant modifications to the manuscript, including adding detailed
descriptions of experiments in legends, the methods and results sections.
Specifically, we have incorporated additional details on how we recorded body
temperature and BAT temperatures, provided proper descriptions of controls,
and included maps of injection sites. Moreover, based on feedback from other
reviewers during the previous review process, we have reorganized the
manuscript and eliminated extraneous details to enhance its readability.

This would include the power analysis that should have been done before the
studies were started indicating the number of animals that should be in each
group; the actual numbers of animals used in each experiment; how they chose
which ones to present in the paper; details about the actual results in the

animals that are included and the ones that are not included, as well as controls;
how they did the statistics; and what the statistical findings were. This is
particularly important for experiments involving stereotaxic injections, some of
which will miss their intended target. How were those cases identified? They
should be analyzed by someone who does not know the physiological results.
The ones that hit the target should be analyzed separately from those that
missed the target, which then serve as anatomical controls. But you need to
present heat maps showing the actual injection placements in both sets of
animals.

R: Again, thank you for bringing up these concerns once again. We would like
to reiterate that we have taken the necessary steps to address these issues by
including the relevant data and statistics in the manuscript. Specifically, we
have listed the numbers in the figures and presented the data in dots to provide
clarity. We have also provided a comprehensive summary of **all numbers and**
**detailed statistics** in **Extended Data Table 2** and highlighted the figures'
details and statistics in the figure legends. Additionally, we ensured that the
experiments were conducted with a minimum of $n = 6$ mice for behavioral
testing, $n = 3$ mice for staining analysis, and $n = 4$ for EMG recording. After the
completion of behavioral tests, we conducted perfusions to check for virus
expression and fiber insertion, and data from mice that showed little or no viral
expression or had a fiber insertion that missed the target (often 0-20%) were
excluded from the analysis (please see **Extended Data Fig. 6, Reviewer only**
**Fig. 4** and methods **lines 69-71** for more information).

**References**

- 1. S. Jung *et al.*, A forebrain neural substrate for behavioral thermoregulation. *Neuron*,
(2021).
- 2. J. N. Flak *et al.*, Leptin-inhibited PBN neurons enhance responses to hypoglycemia in
negative energy balance. *Nature neuroscience* **17**, 1744-1750 (2014).
- 3. W. Z. Yang *et al.*, Parabrachial neuron types categorically encode thermoregulation
variables during heat defense. *Sci Adv* **6**, eabb9414 (2020).
- 4. A. J. Norris, J. R. Shaker, A. L. Cone, I. B. Ndiokho, M. R. Bruchas, Parabrachial
opioidergic projections to preoptic hypothalamus mediate behavioral and physiological
thermal defenses. *eLife* **10**, (2021).
- 5. L. Sun *et al.*, Parabrachial nucleus circuit governs neuropathic pain-like behavior.
*Nature communications* **11**, 5974 (2020).

- 6. D. Mu *et al.*, A central neural circuit for itch sensation. *Science* **357**, 695-699 (2017).
- 7. J. R. Moffitt *et al.*, Molecular, spatial, and functional single-cell profiling of the
hypothalamic preoptic region. *Science* **362**, (2018).
- 8. A. Stengel *et al.*, Central injection of the stable somatostatin analog ODT8-SST induces
a somatostatin2 receptor-mediated orexigenic effect: role of neuropeptide Y and opioid
signaling pathways in rats. *Endocrinology* **151**, 4224-4235 (2010).
- 9. L. Jansky, Neuropeptides and the Central Regulation of Body-Temperature during
Fever and Hibernation. *J Therm Biol* **15**, 329-347 (1990).
- 10. I. Wakabayashi, Y. Tonegawa, T. Shibasaki, Hyperthermic action of somatostatin-28.
*Peptides* **4**, 325-330 (1983).
- 11. W. G. Clark, J. M. Lipton, Brain and pituitary peptides in thermoregulation. *Pharmacol*
*Ther* **22**, 249-297 (1983).
- 12. K. Nakamura, Y. Nakamura, N. Kataoka, A hypothalamomedullary network for
physiological responses to environmental stresses. *Nature reviews. Neuroscience* **23**,
35-52 (2022).
- 13. S. F. Morrison, K. Nakamura, Central Mechanisms for Thermoregulation. *Annual review*
*of physiology* **81**, 285-308 (2019).
- 14. R. A. Pinol *et al.*, Brs3 neurons in the mouse dorsomedial hypothalamus regulate body
temperature, energy expenditure, and heart rate, but not food intake. *Nature*
*neuroscience*, (2018).
- 15. Z. D. Zhao *et al.*, A hypothalamic circuit that controls body temperature. *Proceedings*
*of the National Academy of Sciences of the United States of America* **114**, 2042-2047
(2017).
- 16. C. L. Tan, Z. A. Knight, Regulation of Body Temperature by the Nervous System.
*Neuron* **98**, 31-48 (2018).
- 17. R. A. Pinol *et al.*, Preoptic BRS3 neurons increase body temperature and heart rate via
multiple pathways. *Cell metabolism*, (2021).
- 18. N. Kataoka, Y. Shima, K. Nakajima, K. Nakamura, A central master driver of
psychosocial stress responses in the rat. *Science* **367**, 1105-1112 (2020).

REVIEWER COMMENTS

Reviewer #1 (Remarks to the Author):

The authors have carefully addressed the critiques I commented on in the first round of reviews. This comprehensive study shows that the PBN-to-DMH work in parallel with PBN-to-POA in regulating body temperature. A large amount of data are presented. I don't have any further criticism that I want to raise. Congratulations on accomplishing such an excellent study.

Reviewer #2 (Remarks to the Author):

We appreciate the authors carefully addressed many of our concerns. However, several important questions have not been answered.

In our major point 1, I asked 'What is the relationship between neurons in DMH and MPO that received inputs from LPB?'. The authors said that 'DMH neurons, including

LPB-innervating DMH neurons, are expected to act downstream of LPB

innervating MnPO/VMPO neurons to form a feed-forward pathway in warm defense.'. It will be great for the authors to provide some experimental results to support their claim.

In our major point 2, I asked 'why both DMHVglut2 vs. DMHVgat are required for cold induced defensive behavior?' The authors again proposed a possibility 'we proposed that DMHVglut2 neurons primarily provide excitatory input to premotor neurons in the RPa to stimulate thermogenesis, while DMHVgat neurons may mainly suppress POA hypothalamic neurons to inhibit heat loss.' but without data to support this hypothesis. Is DMHVglut2 neurons to RPa really stimulate thermogenesis?

In our major point 5, I like to see the sequencing results been validated with RNAscope staining to determine what is the best marker for the DMH-projecting LPBVglut2 neurons that is important for cold.

REVIEWER COMMENTS

Reviewer #1 (Remarks to the Author):

The authors have carefully addressed the critiques I commented on in the first round of reviews. This comprehensive study shows that the PBN-to-DMH work in parallel with PBN-to-POA in regulating body temperature. A large amount of data are presented. I don't have any further criticism that I want to raise. Congratulations on accomplishing such an excellent study.

R: We appreciate the recognition from this reviewer.

Reviewer #2 (Remarks to the Author):

We appreciate the authors carefully addressed many of our concerns. However, several important questions have not been answered.

In our major point 1, I asked 'What is the relationship between neurons in DMH and MPO that received inputs from LPB?'. The authors said that 'DMH neurons, including LPB-innervating DMH neurons, are expected to act downstream of LPB innervating MnPO/VMPO neurons to form a feed-forward pathway in warm defense.'. It will be great for the authors to provide some experimental results to support their claim.

R: In an effort to substantiate our claims, we undertook patch-clamp electrophysiology tests. In order to ascertain whether there is LPB innervating POA neurons direct innervate LPB innervating DMH neurons, we injected AAV1-hSyn-Cre into the LPB, and AAVs carrying Cre-dependent Channelrhodopsin-2 (AAV9-hSyn-DIO-ChR2-EYFP) into the POA and Cre-dependent mCherry (AAV9-DIO-mCherry) into the DMH of C57BL/6J mice. Light-induced excitatory postsynaptic currents (EPSCs) of DMH neurons were recorded through patch-clamp while photostimulating ChR2-expressing neural terminals in the DMH projected from LPB innervating POA neurons (**Reviewer only Fig. 1a**). Light stimulations faithfully induced EPSCs, which was found within the range of a monosynaptic connection (**Reviewer only Fig. 1b-c**). Around 15% of the recorded DMH neurons (3 of 20 neurons selected at random) exhibited EPSCs in response to light stimulations (**Reviewer only Fig. 1d**). These data show that the LPB-innervating DMH neurons receive inputs from the LPB-innervating POA neurons, suggesting that LPB-innervating DMH neurons might act downstream of LPB-innervating POA neurons to form a feed-forward pathway.

We did not directly test whether this pathway is involved in feed-forward warm defense since it requires tons of evidence to resolve the complications. Originally, there are many evidences showing a GABAergic POA-DMH projection to inhibit thermogenesis (1-3). At the same time, more evidences point out that glutamatergic POA-DMH projection may help to reduce T_{core} . For example, most BDNF neurons are glutamatergic and projects to DMH (1, 4). Glutamatergic QRFP neurons rely on DMH neurons to induce torpor (5). Therefore, given these complications, although we believe that LPB-innervating DMH neurons might act downstream of LPB-innervating POA neurons to control warm defense, it is beyond the scope of the current study. We hope the reviewer will understand the situation. Therefore, we did not include the data in the manuscript and would like to perform more careful analysis on this issue in the future.

Reviewer only Fig. 1 | (a) Scheme for patch clamp recording LPB innervating DMH neurons after activating the terminal of LPB innervating POA neurons. To do so, we simultaneously injected AAV1-hSyn-Cre into the LPB, and AAVs carrying Cre-dependent Channelrhodopsin-2 (AAV9-hSyn-DIO-ChR2-EYFP) into the POA and Cre-dependent mCherry (AAV9-DIO-mCherry) into the DMH in C57BL/6J mice. (b) Induction of EPSCs in DMH neurons by light stimulation of neural terminals in the DMH projected from LPB innervating POA neurons (c) The mean latency of the induced EPSC after photoactivation of responsive neurons. (n = 20 trials from 3 neurons). (d) The EPSC response rate of DMH neurons to light stimulation. A total of 3 responsive neurons out of 20 randomly recorded DMH neurons from 3 mice.

In our major point 2, I asked 'why both $DMH^{V_{glut2}}$ vs. $DMH^{V_{gat}}$ are required for cold induced defensive behavior?' The authors again proposed a possibility 'we proposed that $DMH^{V_{glut2}}$ neurons primarily provide excitatory input to premotor neurons in the RPa to stimulate thermogenesis, while $DMH^{V_{gat}}$ neurons may mainly suppress POA hypothalamic neurons to inhibit heat loss.' but without data to support this hypothesis. Is $DMH^{V_{glut2}}$ neurons to RPa really stimulate thermogenesis?

R: The glutamatergic DMH-rMR/RPa projection in promoting thermogenesis has been shown before in rats and mice repetitively. For example, Kataoka et al previously determined glutamatergic DMH-rMR projection drives BAT thermogenesis in rat (6). As cited figures shown below, activation of cell bodies in the DMH by **optogenetics** and following a saline nanoinjection into the rMR (**Cited Fig. 1B and 1C**) elicited increases in BAT SNA and T_{BAT} (**Cited Fig. 1D**). In contrast, following a subsequent nanoinjection of glutamate receptor antagonist AP5/CNQX into the rMR (**Cited Fig. 1B and 1C**), neither BAT SNA nor T_{BAT} was increased by activation of DMH neurons (**Cited Fig. 1E**).

Machado et al also identified DHA^{Vglut2}-RPa pathway mediate BAT thermogenesis in mice (7). They found optogenetic inhibition of the axon terminals of ArchT-GFP expressing DMH^{Vglut2} neurons in the RPa caused a significant reduction of baseline Tc (Cited Fig. 2A and 2B).

Cited Fig. 1 | (A) In vivo experiment to examine the effect of antagonizing glutamate receptors in the rMR on physiological responses to photostimulation of DMH neurons (DMH cell body*). (B and C) Nanoinjection sites in the rMR are mapped in (B). Each circle indicates a site of saline and AP5/CNQX injections made at the same location in each rat. The effect of saline was always tested first. Each injection site was labeled with fluorescent microspheres (arrow in [C]). Scale bar, 500 μ m. (D and E) Effect of illumination of ChIEF-tdTomato-expressing cells in the DMH on BAT thermogenic and cardiovascular activities following saline (D) or AP5/ CNQX injection (E) into the rMR. Results from the same rat are shown. Horizontal bars, 30 s.

Cited Fig. 2 | (A) A schematic figure of the protocol for inhibition of DHA VGLUT2+ fibers in the RPa using AAV-DIO-ArchT-GFP. (B) T_c measurement in Vglut2-IRES-cre mice injected in the DHA bilaterally with AAV-DIO-ArchT-GFP (n = 5) or controls injected with GFP (n = 6) during optogenetic inhibition of DHA^{Vglut2} terminals in the RPa (35.73 °C ± 0.02°C ArchT versus 36.15 °C ± 0.008 °C GFP controls during first 30 min after initiation of laser stimulation; Mann-Whitney test; ± SEM; *p < 0.0001). The laser inhibition causes an ~1 °C fall in T_c at maximum, which is about 10 min after the termination of the laser inhibition.

Nevertheless, we directly determined whether activation of the glutamatergic DMH-RPa projection would promote thermogenesis in mice. We expressed Cre-dependent ChR2 in DMH of the Vglut2-ires-Cre mice and activated the axon terminals of DMH^{Vglut2} neurons in the RPa by blue laser (**Extended Data Fig. 10a**). Compared with the GFP control mice, activation of the terminals of ChR2-expressing DMH^{Vglut2} neurons in the RPa significantly increased T_{core}, yet it only slightly increased the physical activity at the beginning of laser stimulation (**Extended Data Fig. 10b,c**). Taken together, the DMH^{Vglut2}-RPa projection is sufficiently to increase thermogenesis in mice. We added these data in **Extended Data Fig. 10a-c** and modified manuscript accordingly (lines 533-535).

Extended Data Fig. 10 | (a) Activating the DMH^{Vglut2}→RPa projection via optogenetic activation of DMH^{Vglut2 & ChR2} neural terminals in the RPa. AAVs carrying Cre-dependent ChR2 (AAV9-hEF1a-DIO-hChR2-EYFP) were injected into the DMH of Vglut2-IRES-Cre mice. An optical fiber was implanted above the RPa and used for optogenetic activation

of neural terminals. AAV9-hSyn-Flex-GFP was used as the control. (b-c) Changes of T_{core} (b) and physical activity (c) after photoactivation of DMH^{Vglut2 & Chr2} neural terminals in the RPa (Chr2, n = 9 mice; GFP, n = 6 mice). Light pattern: 473 nm, 12 mW, 10 Hz, 10 ms, 2-s on followed by 2-s off, with the cycles repeating for 30 min.

In our major point 5, I like to see the sequencing results been validated with RNAscope staining to determining what is the best marker for the DMH-projecting LPB^{Vglut2} neurons that is important for cold.

R: Thanks for the reviewer's interest and curiosity, we now present more data as this reviewer suggested. As we previously stated, there were few Gal-Cre⁺ and TH⁺ neurons in the LPB. Thus, we validated the others: N4bp2, Spint2, CCK, GRP and FoxP2 (in the order of enrichment score). However, N4bp2 was not specifically expressed in the LPB according to Allen database (**Reviewer only Fig. 2a**). Spint2 appeared to enrich in the LPB according to Allen database (**Reviewer only Fig. 2b**). Yet, no antibodies or RNAscope probes available currently. The newly synthesized probe would take 2-3 months to synthesize and it may take even longer to ship from US to China. Therefore, we prioritized the validation of CCK, GRP, and FoxP2.

These results are shown below: about 38% of DMH-projecting LPB^{CCK} neurons were sensitive to cold exposure, accounting for 20% of cold-activated LPB neurons (**Reviewer only Fig. 2c**). And nearly 30% of DMH-projecting LPB^{GRP} neurons were sensitive to cold exposure, accounting for 15% of cold-activated LPB neurons (**Reviewer only Fig. 2d,e**). About 60% of DMH-projecting LPB^{FoxP2} neurons (shown as FoxP2⁺CTB555⁺ (DMH-projecting)) were sensitive to cold exposure, accounting for 32% of cold-activated LPB neurons (**Reviewer only Fig. 2f**). Therefore, FoxP2 might be another good marker for the DMH-projecting LPB^{Vglut2} neurons. However, due to the lack of FoxP2-Cre mice, we have not been able to test the functionality of the LPB^{FoxP2}-DMH projection at this time. Thus, we didn't present these data, and would like to further test it in the future.

Reviewer only Fig. 2 | (a-b) ISH data of N4bp2 (a) and Spint2 (b) from Allen brain. (c) Overlap between DMH-projecting LPB^{CCK} neurons and cold-induced cFos (n = 2 mice). To label DMH-projecting LPB^{CCK} neurons, we injected retrograde AAVs carrying Cre-dependent GFPL10 (AAV-Retro-CAG-Flex-GFPL10) in the DMH of CCK-IRES-Cre mice, which drove the expression of GFPL10 in the LPB. (d) Overlap between LPB^{GRP}

neurons (labeled with GRP-Cre & Ai14) and cold-induced cFos (n = 3 mice). (e) Overlap between DMH-projecting LPB^{GRP} neurons and cold-induced cFos (n = 3 mice). To label DMH-projecting LPB^{GRP} neurons, we injected retrograde AAVs carrying Cre-dependent GFPL10 (AAV-Retro-CAG-Flex-GFPL10) in the DMH of GRP-IRES-Cre mice, which drove the expression of GFPL10 in the LPB. (e) Overlap between DMH-projecting LPB^{FoxP2} neurons and cold-induced cFos (n = 3 mice). To label DMH-projecting LPB^{FoxP2} neurons, we injected retrograde CTB555 in the DMH of C57 mice, then stained the FoxP2 in the LPB. Merged cells were indicated by white arrows. Scale bar, 100µm. LPBel, lateral parabrachial nucleus, external and lateral part.

Reference:

1. Z. D. Zhao *et al.*, A hypothalamic circuit that controls body temperature. *Proceedings of the National Academy of Sciences of the United States of America* **114**, 2042-2047 (2017).
2. C. L. Tan *et al.*, Warm-Sensitive Neurons that Control Body Temperature. *Cell* **167**, 47-+ (2016).
3. S. F. Morrison, K. Nakamura, Central neural pathways for thermoregulation. *Frontiers in bioscience* **16**, 74-104 (2011).
4. C. L. Tan, Z. A. Knight, Regulation of Body Temperature by the Nervous System. *Neuron* **98**, 31-48 (2018).
5. T. M. Takahashi *et al.*, A discrete neuronal circuit induces a hibernation-like state in rodents. *Nature*, (2020).
6. N. Kataoka, H. Hioki, T. Kaneko, K. Nakamura, Psychological stress activates a dorsomedial hypothalamus-medullary raphe circuit driving brown adipose tissue thermogenesis and hyperthermia. *Cell Metab* **20**, 346-358 (2014).
7. N. L. S. Machado *et al.*, A Glutamatergic Hypothalamomedullary Circuit Mediates Thermogenesis, but Not Heat Conservation, during Stress-Induced Hyperthermia. *Current biology : CB* **28**, 2291-2301 e2295 (2018).

REVIEWERS' COMMENTS

Reviewer #2 (Remarks to the Author):

The authors successfully addressed my remaining concerns with new data and analysis. I agree with the authors on not including some of the preliminary results in this current manuscript, but I hope these new results could potentially lead to exciting discoveries in near future.

REVIEWERS' COMMENTS

Reviewer #2 (Remarks to the Author):

The authors successfully addressed my remaining concerns with new data and analysis. I agree with the authors on not including some of the preliminary results in this current manuscript, but I hope these new results could potentially lead to exciting discoveries in near future.

R: We are grateful for this reviewer's recognition and happy to do something that could lead to new exciting discoveries in near future.